# Emulator-based Bayesian inference on non-proportional scintillation models by compton-edge probing

David Breitenmoser [1,2] ✉, Francesco Cerutti[3], Gernot Butterweck[1], Malgorzata Magdalena Kasprzak [1] & Sabine Mayer [1]

Scintillator detector response modeling has become an essential tool in various research fields such as particle and nuclear physics, astronomy or geophysics. Yet, due to the system complexity and the requirement for accurate electron response measurements, model inference and calibration remains a challenge. Here, we propose Compton edge probing to perform non-proportional scintillation model (NPSM) inference for inorganic scintillators. We use laboratory-based gamma-ray radiation measurements with a NaI(Tl) scintillator to perform Bayesian inference on a NPSM. Further, we apply machine learning to emulate the detector response obtained by Monte Carlo simulations. We show that the proposed methodology successfully constrains the NPSM and hereby quantifies the intrinsic resolution. Moreover, using the trained emulators, we can predict the spectral Compton edge dynamics as a function of the parameterized scintillation mechanisms. The presented framework offers a simple way to infer NPSMs for any inorganic scintillator without the need for additional electron response measurements.

Inorganic scintillation detectors are a prevalent tool to measure ionizing radiation in various research fields such as nuclear and particle physics, astronomy, or planetary science[1–7]. Other applications include radiation protection, medical diagnostics, and homeland security[8,9]. In almost all applications, the measured signal needs to be deconvolved to infer the properties of interest, e.g. the flux from a gamma-ray burst or the elemental composition on a comet. This deconvolution requires accurate detector response models and consequently detailed knowledge about the scintillation mechanisms themselves.

Detector response models can either be derived empirically by radiation measurements or numerically using Monte Carlo simulations[10]. Regarding the numerical derivation, the most common approach to simulate the detector response is to use a proportional energy deposition model. In this model, the scintillation light yield is assumed to be proportional to the deposited energy[6,11]. Consequently,

the detector response characterization is reduced to a comparably simple energy deposition problem, which can be solved by any standard multi-purpose Monte Carlo code.

However, thanks to the development of the Compton coincidence measurement technique[12], recent studies could conclusively confirm the conjecture reported in earlier investigations[13–15] that not only organic but also inorganic scintillators exhibit a pronounced non-proportional relation between the deposited energy and the scintillation light yield[16–18]. The origin of this scintillation non-proportionality seems to be linked to the intrinsic scintillation response to electrons and the different mechanisms associated with the creation and transport of excitation carriers in the scintillation crystal[19,20]. Nonetheless, our understanding of these phenomena is still far from complete and, thanks to the advent of advanced experimental techniques and the development of new scintillator materials, interest in scintillation physics has steadily grown over the past years[16–24].

[1]Department of Radiation Safety and Security, Paul Scherrer Institute (PSI), Forschungsstrasse 111, Villigen PSI 5232, Switzerland. [2]Department of Physics, Swiss Federal Institute of Technology (ETH), Otto-Stern-Weg 5, Zurich 8093, Switzerland. [3]European Organization for Nuclear Research (CERN), Esplanade des Particules 1, Geneva 1211, Switzerland. ✉e-mail: david.breitenmoser@psi.ch

Regarding the detector response modeling, the scintillation non-proportionality has two major implications. First, it leads to an intrinsic spectral broadening and thereby sets a lower limit on the spectral resolution achievable with the corresponding scintillator[1,25–28]. Second, various studies stated the conjecture that specific spectral features such as the Compton edges are shifted and distorted as a result of the non-proportional scintillation response[1,14,15,29,30]. Furthermore, additional studies revealed a complex dependence of the scintillation non-proportionality on various scintillator properties including the activator concentration, the temperature, and the crystal size, among others[1,21,22,25,28,31–34].

Based on these findings, we conclude that non-proportional scintillation models (NPSM) should be included in the detector response simulations to prevent systematic errors in the predicted spectral response. Non-proportional effects are known to increase with increasing crystal size[25,28,31]. NPSMs are therefore particularly relevant for scintillators with large crystal volumes, e.g. in dark matter research, total absorption spectroscopy or remote sensing[1–7,30]. In addition, especially due to the sensitivity of the activator concentration and impurities[34], NPSMs need to be calibrated for each individual detector system. In case the scintillator properties change after detector deployment, e.g. due to radiation damage or temperature changes in space, this calibration should be repeated regularly.

Currently, K-dip spectroscopy, the already mentioned Compton coincidence technique as well as electron beam measurements are the only available methods to calibrate NPSM[12,35–38]. Moreover, only a very limited number of laboratories are able to perform these measurements. Therefore, these methods are not readily available for extensive calibration campaigns of custom detectors, e.g. large satellite probes or scintillators for dark matter research. Additionally, they cannot be applied during detector deployment, which, as discussed above, might be important for certain applications such as deep space missions.

In this study, we propose Compton edge probing together with Bayesian inversion to infer and calibrate NPSMs. This approach is motivated by the already mentioned conjecture, that the Compton edge shifts as a result of the scintillation non-proportionality[1,14,15,29,30]. We obtain the spectral Compton edge data by gamma-ray spectrometry using a NaI(Tl) scintillator and calibrated radionuclide sources for photon irradiations under laboratory conditions. We apply Bayesian inversion with state-of-the-art Markov-Chain Monte Carlo algorithms[39] to perform the NPSM inference with the gamma-ray spectral data. In contrast to traditional frequentist methods or simple data-driven optimization algorithms, a Bayesian approach offers a natural, consistent, and transparent way of combining prior information with empirical data to infer scientific model properties using a solid decision theory framework[40–42]. We simulate the detector response using a multi-purpose Monte Carlo radiation transport code in combination with parallel computing. To meet the required evaluation speed for the Bayesian inversion solver, we use machine learning trained polynomial chaos expansion (PCE) surrogate models to emulate the simulated detector response[43]. This approach offers not only a simple way to calibrate NPSMs with minimal effort—especially during the detector deployment—but it also allows additional insights into the non-proportional scintillation physics without the need for additional electron response measurements.

## Results

### Compton edge probing

To obtain the spectral Compton edge data, we performed gamma-ray spectrometry under controlled laboratory conditions[30]. The adopted spectrometer consisted of four 10.2 cm × 10.2 cm × 40.6 cm prismatic NaI(Tl) scintillation crystals with individual read-out. We used seven different calibrated radionuclide sources ($^{57}$Co, $^{60}$Co, $^{88}$Y, $^{109}$Cd, $^{133}$Ba, $^{137}$Cs, and $^{152}$Eu) for the radiation measurements. However, only $^{60}$Co, $^{88}$Y, and $^{137}$Cs could be used for

Compton edge probing. For the remaining sources, the Compton edges were obscured by additional full energy peaks (FEPs) and associated Compton continua. We used those remaining sources for energy and resolution calibrations. A schematic depiction of the measurement setup is shown in Fig. 1a.

### Forward modeling

We simulated the detector response for the performed radiation measurements using the multi-purpose Monte Carlo code FLUKA[44]. The performed simulations feature fully coupled photon, electron, and positron radiation transport for our source-detector configuration with a lower kinetic energy threshold of 1 keV. As shown in Fig. 1a, the applied mass model includes all relevant detector and source components in high detail. On the other hand, the laboratory room together with additional instruments and equipment are modeled in less detail. For this simplification, care was taken to preserve the overall opacity as well as the mass density.

We used a mechanistic model recently published by Payne and his co-workers to include the non-proportional scintillation physics in our simulations[17,18,22]. In general, the sequence of scintillation processes in inorganic scintillators can be qualitatively divided into five steps[20,45,46]. After interaction of the ionizing radiation with the scintillator, the emitted high-energetic electrons are relaxed by the production of numerous secondary electrons, phonons, and plasmons. The low energetic secondary electrons are then thermalized by a phonon coupling mechanism producing excitation carriers, i.e. electron–hole pairs ($e^-$/h) and excitons. These excitation carriers are then transferred to the luminescent centers within the scintillation crystal, where they recombine and induce radiative relaxation of the excited luminescent centers producing scintillation photons. The first two processes, i.e. the interaction of the ionizing radiation with the scintillator as well as the $e^-$–$e^-$ relaxation, are explicitly simulated by the Monte Carlo code. The creation and migration of the excitation carriers on the other hand is accounted for by Payne's mechanistic model.

In this mechanistic model it is assumed that only excitons are capable of radiatively recombine at the luminescent centers. Consequently, $e^-$/h pairs need to convert to excitons by the classic Onsager mechanism[47] in order to contribute to the scintillation emission. In addition, creation and migration of the excitation carriers compete with several quenching phenomena. The quenching mechanisms considered in Payne's model are the trapping of $e^-$/h pairs at point defects[20,22] as well as exciton–exciton annihilation described by the Birks mechanism[48].

Using this NPSM, the non-proportional light yield $L$ as a function of the differential energy loss $dE$ per differential path length $ds$ for electrons is given by[22]

$$L(dE/ds) \propto \frac{1 - \eta_{e/h} \exp\left[-\frac{dE/ds}{dE/ds|_{\text{Ons}}} \exp\left(-\frac{dE/ds|_{\text{Trap}}}{dE/ds}\right)\right]}{1 + \frac{dE/ds}{dE/ds|_{\text{Birks}}}} \quad (1)$$

where $\eta_{e/h}$, $dE/ds|_{\text{Ons}}$, $dE/ds|_{\text{Trap}}$ and $dE/ds|_{\text{Birks}}$ are the model parameters characterizing the fraction of excitation carriers, which are created as $e^-$/h pairs at the thermalization phase, as well as the stopping power related to the Onsager, trapping and Birks mechanisms, respectively. As a result, all the parameters of the NPSM reflect physical processes after thermalization of the secondary particles, i.e. generation and transport of excitation carriers. Consequently, these processes and thereby also the corresponding parameters can be regarded as statistically independent with respect to the energy of the secondary particles. From a physics perspective, it is important to note that the Onsager and trapping mechanisms are coupled in a nonlinear way, whereas the Birks mechanism can be regarded as independent of the other mechanisms. As discussed in detail by Vasil'ev and Gektin[20], we may therefore interpret the trapping of $e^-$/h pairs as a screening

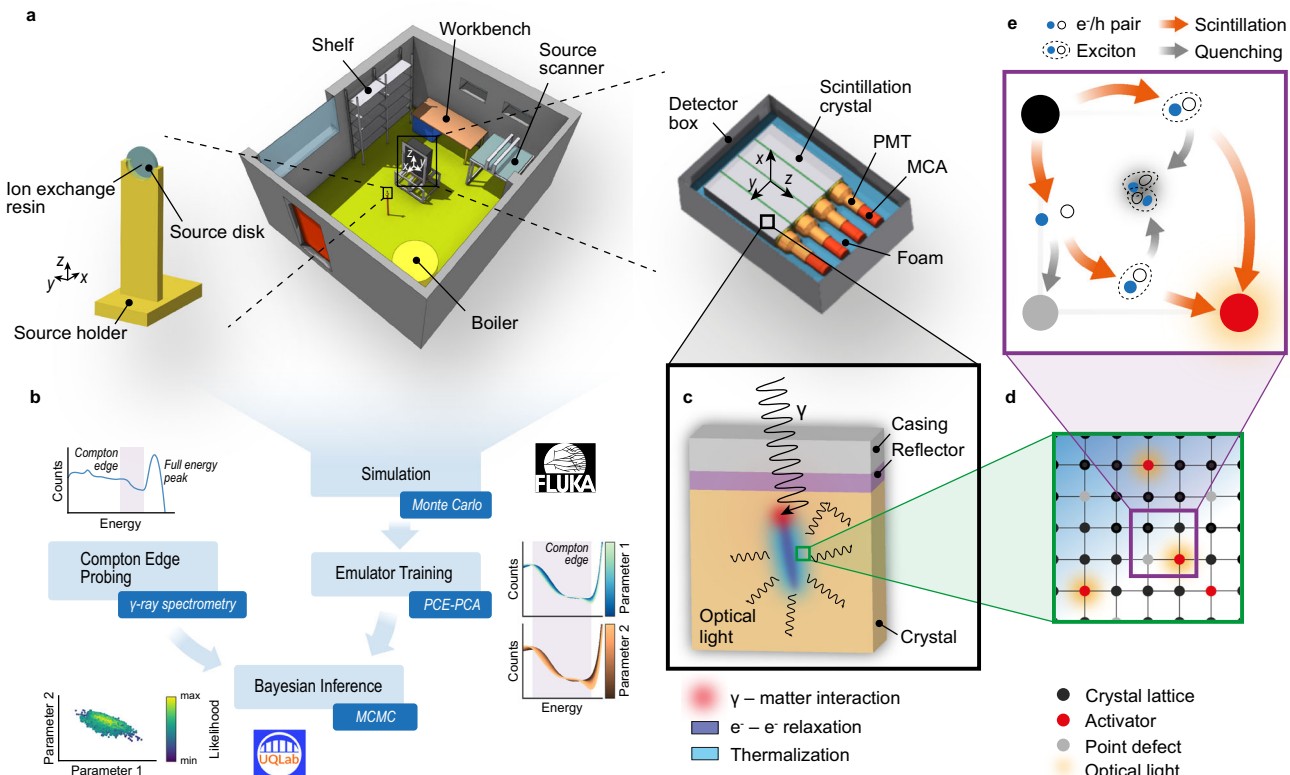

**Fig. 1 | Compton edge probing to perform Bayesian inference on non-proportional scintillation models. a** Monte Carlo mass model of the experimental setup to perform Compton edge probing with an inorganic gamma-ray scintillation spectrometer under laboratory conditions. The spectrometer consists of four individual 10.2 cm × 10.2 cm × 40.6 cm prismatic NaI(Tl) scintillation crystals with the associated photomultiplier tubes (PMT), the electronic components, e.g. the multi-channel analyzers (MCA), embedded in a thermal-insulating and vibration-damping polyethylene (PE) foam protected by a rugged aluminum detector box. We inserted radiation sources consisting of a radionuclide carrying ion exchange sphere (diameter 1 mm) embedded in a 25 mm × 3 mm solid plastic disc into a custom low absorption source holder made out of a polylactide polymer (PLA) and placed this holder on a tripod in a fixed distance of 1 m to the detector front on the central detector $x$-axis. The mass model figures were created using the

graphical interface FLAIR[63]. For better visibility and interpretability, we applied false colors. **b** Overview of the Bayesian inference framework highlighting the gamma-ray spectrometry based Compton edge probing measurements, the Monte Carlo simulations using the multi-purpose code FLUKA[44] combined with the machine learning trained polynomial chaos expansion (PCE) emulator models supported by principal component analysis (PCA) as well as the Bayesian inference by Markov Chain Monte Carlo (MCMC) itself using UQLab[72]. **c** Radiation transport mechanisms inside the inorganic scintillation crystal, which is surrounded by a thin reflector layer and a rugged aluminum crystal casing. **d** Schematic representation of an inorganic scintillation crystal lattice including the activator atoms and point defects. **e** Mechanistic depictions of the various scintillation and quenching pathways for electron-hole pairs (e⁻/h) as well as excitons within the inorganic scintillation crystal lattice. Adapted from ref. 30.

mechanism on the Onsager term in Eq. (1). A scheme highlighting the individual scintillation processes included in the present study is presented in the Fig. 1c–e.

## Bayesian inversion

We applied Bayesian inversion using Markov Chain Monte Carlo[39] to infer the NPSM parameters as well as to predict spectral and resolution scintillator properties from the measured Compton edge spectra and our forward model. To account for the sensitivity of the NPSMs on the activator concentration and other scintillation crystal-specific properties, we developed two separate inversion pipelines. In the first approach, Bayesian inversion is carried out separately for each of the four crystals, using their individual pulse-height spectra. In the second pipeline, we consider all four scintillation crystals as one integrated scintillator and perform the Bayesian inversion on the combined pulse-height spectra (sum channel). Subsequently, we will refer to these two approaches as the *single* and *sum* mode inversion pipelines. For both pipelines, we performed the Bayesian inversion on the $^{60}$Co (activity $A = 3.08(5) \times 10^5$ Bq) spectral dataset[30] leaving the remaining measurements for validation. $^{60}$Co possesses two main photon emission lines at 1173.228(3) keV and 1332.492(4) keV with corresponding Compton edges according to the Compton scattering theory (Methods) at 963.419(3) keV and 1118.101(4) keV, respectively. However, in

this study, we will focus on the lower Compton edge at 963.419(3) keV, because the upper edge is heavily obscured by the FEP at 1173.228(3) keV. Furthermore, as suggested by previous investigators[18,22], we fixed the Onsager-related stopping power parameter $dE/ds|_{\text{Ons}}$ to 36.4 MeV cm$^{-1}$ in both pipelines.

Because the high-fidelity radiation transport simulations described in the previous section are computationally intense, we emulated the detector response as a function of the NPSM parameters using a machine learning trained vector-valued PCE surrogate model[43]. The surrogate model has excellent evaluation speed $\mathcal{O}(10^{-4}\,\text{s})$ on a local workstation compared to $\mathcal{O}(10^3\,\text{s})$ required for a single Monte Carlo simulation with sufficient precision on a computer cluster. Correspondingly, the surrogate model provides a significant acceleration of our Bayesian inversion computations, reducing their processing time by a factor of $10^7$. Considering the minimum number of forward model evaluations needed for a single Markov chain (Methods), the evaluation time can be reduced from $\mathcal{O}(10^2\,\text{d})$ to $\mathcal{O}(1\,\text{s})$.

Following the sum mode inversion pipeline, we present the solution to our inversion problem as a multivariate posterior distribution estimate in Fig. 2. We find a unimodal solution with a maximum a posteriori (MAP) probability estimate given by $\eta_{\text{e/h}} = 5.96^{+0.10}_{-0.17} \times 10^{-1}$, $dE/ds|_{\text{Trap}} = 1.46^{+0.03}_{-0.31} \times 10^1$ MeV cm$^{-1}$ and $dE/ds|_{\text{Birks}} = 3.22^{+0.46}_{-0.44} \times 10^2$ MeV cm$^{-1}$, where we used the central

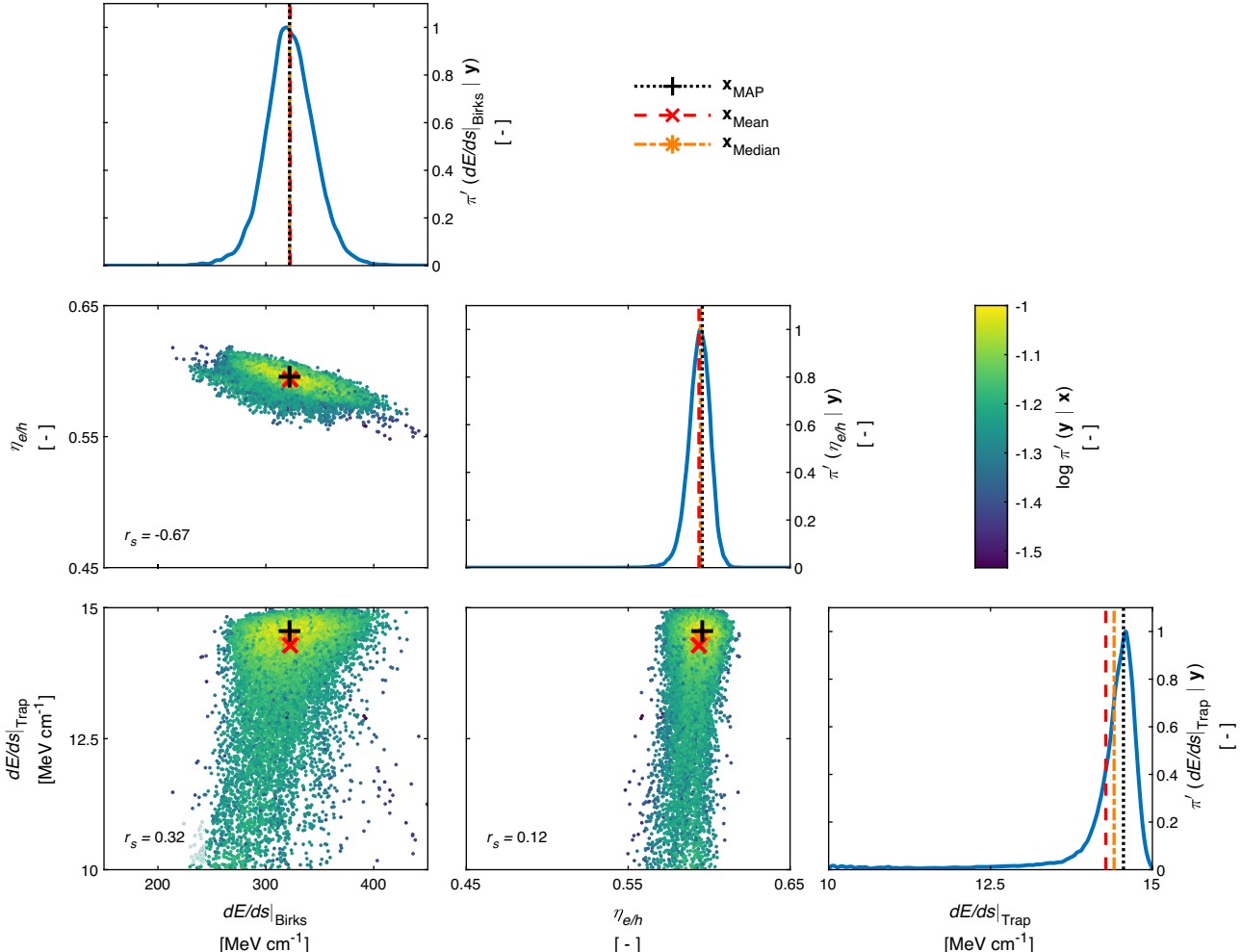

**Fig. 2 | Posterior distribution estimate.** As a result of the sum mode inversion pipeline, the off-diagonal subfigures present samples from the multivariate posterior marginals given the experimental dataset $\mathbf{y}$ for the model parameters $\mathbf{x} := (dE/ds|_{Birks}, \eta_{e/h}, dE/ds|_{Trap})^\top$. We colored these samples by the corresponding normalized multivariate log-likelihood function values $\log \pi'(\mathbf{y}|\mathbf{x})$. In addition, the Spearman's rank correlation coefficient $r_s$ is provided for the model parameters in the corresponding off-diagonal subfigures. The subfigures on the diagonal axis highlight the normalized univariate marginal likelihood $\pi'(x|\mathbf{y})$ for the model parameter $x$. Both, the univariate and multivariate likelihood values, were normalized by their corresponding global maxima. Derived posterior point estimators, i.e. the maximum a posteriori (MAP) probability estimate $\mathbf{x}_{MAP}$, the posterior mean $\mathbf{x}_{Mean}$ and the posterior median $\mathbf{x}_{Median}$, are indicated as well in each subfigure.

credible intervals with a probability mass of 95% to estimate the associated uncertainties. Combining the individual multivariate posterior distribution estimates from the single mode inversion pipeline, we obtain statistically consistent estimates, i.e. $\eta_{e/h} = 5.87^{+0.24}_{-0.20} \times 10^{-1}$, $dE/ds|_{Trap} = 1.41^{+0.17}_{-0.15} \times 10^{1}$ MeV cm$^{-1}$ and $dE/ds|_{Birks} = 3.17^{+1.11}_{-0.82} \times 10^{2}$ MeV cm$^{-1}$.

It is worth noting that, considering the uncertainty estimates, we observe only minor differences between the different posterior point estimators for both inversion pipelines (Fig. 2 and Supplementary Figs. S11–S14). However, we find statistically significant differences between the posterior point estimators for the individual scintillation crystals (Supplementary Table S2). Furthermore, our results significantly differ from best-estimate literature values, which we obtained using linear temperature interpolation on a dataset provided by Payne and his co-workers, i.e. $\eta_{e/h} = 4.53 \times 10^{-1}$, $dE/ds|_{Trap} = 1.2 \times 10^{1}$ MeV cm$^{-1}$ and $dE/ds|_{Birks} = 1.853 \times 10^{2}$ MeV cm$^{-1}$ for an ambient temperature of 18.8 °C[22].

## Compton edge predictions

We can use the trained PCE surrogate models to predict the spectral Compton edge as a function of the NPSM parameters and consequently the parameterized scintillation and quenching phenomena. In Fig. 3a–c, we present the spectral response of the PCE surrogate model

for the sum channel as a function of the Birks-related stopping power parameter $dE/ds|_{Birks}$, the free carrier fraction $\eta_{e/h}$ and the trapping-related stopping power parameter $dE/ds|_{Trap}$. We observe a shift of the Compton edge toward smaller spectral energies for an increase in $dE/ds|_{Birks}$ and $\eta_{e/h}$ as well as a decrease in $dE/ds|_{Trap}$.

We leveraged the analytical relation between the polynomial chaos expansion and the Hoeffding-Sobol decomposition[49] to perform a global sensitivity analysis of the NPSM. Using the sum mode inversion pipeline, we present total Sobol indices $S^T$ for the model parameters $dE/ds|_{Birks}$, $\eta_{e/h}$ and $dE/ds|_{Trap}$ in Fig. 3e. We find that the total Sobol indices can be ordered as $S^T(\eta_{e/h}) > S^T(dE/ds|_{Birks}) > S^T(dE/ds|_{Trap})$ over the entire spectral Compton edge domain indicating a corresponding contribution to the total model response variance. We get consistent results for a Hoeffding-Sobol sensitivity analysis of the individual scintillation crystals (Supplementary Fig. S17).

In addition, we can also predict the spectral Compton edge using the prior and posterior predictive density estimates obtained by the two inversion pipelines. A comparison of these densities for the sum mode inversion pipeline indicates that our methodology successfully constrains the adopted NPSM (Fig. 3d). However, we find also some model discrepancies, especially around the Compton continuum at the very low end of the investigated spectral range (< 920 keV). We get

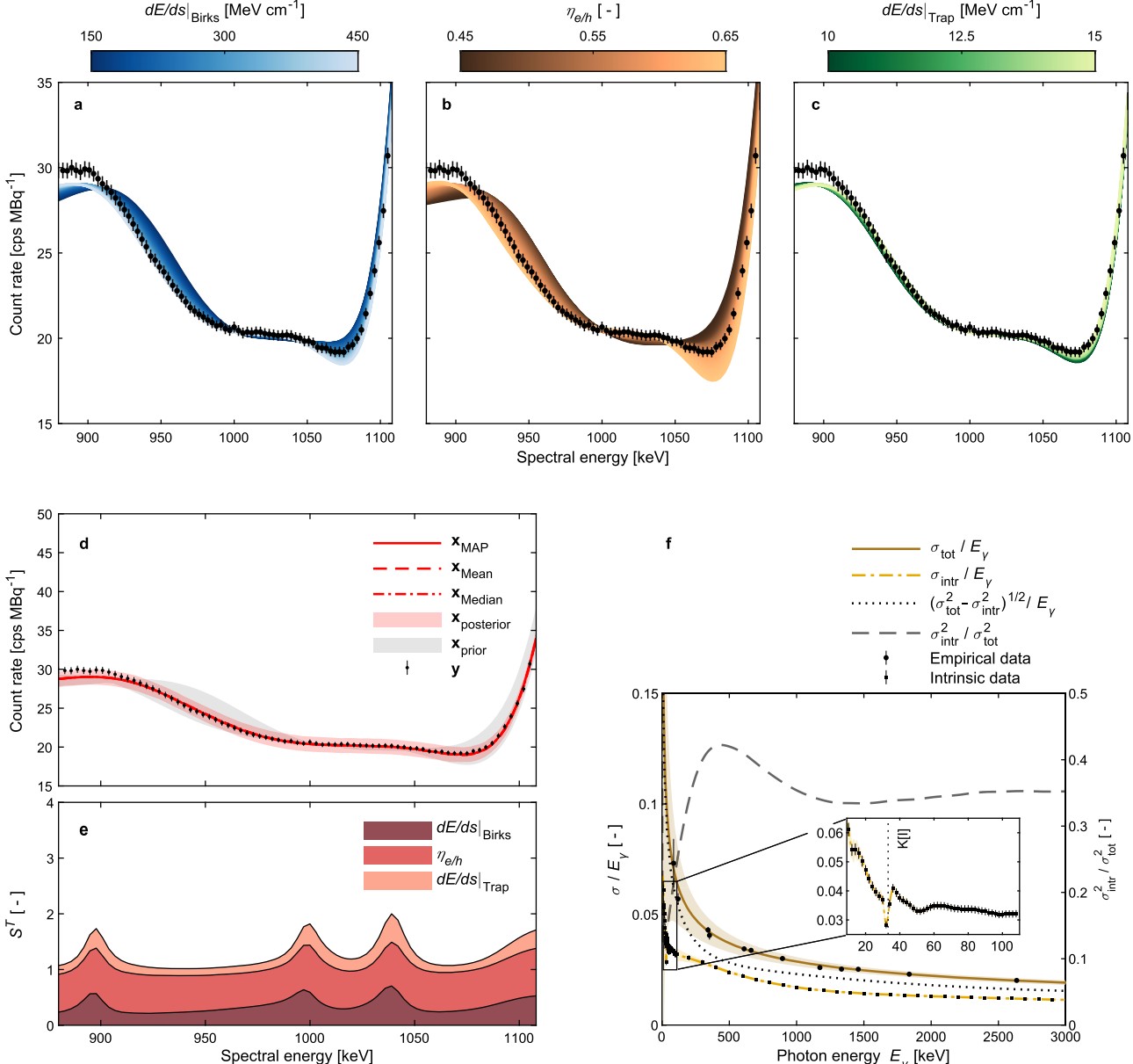

**Fig. 3 | Compton edge and intrinsic resolution predictions. a–c** Compton edge dynamics characterized by the trained polynomial chaos expansion emulator as a function of the individual non-proportional scintillation model parameters, i.e. the Birks related stopping power parameter $dE/ds|_{Birks}$, the free carrier fraction $\eta_{e/h}$ as well as the trapping related stopping power parameter $dE/ds|_{Trap}$, for the sum channel and the corresponding prior range. We fixed the remaining parameters at the corresponding maximum a posteriori (MAP) probability estimate values $\mathbf{x}_{MAP}$. The experimental data $\mathbf{y}$ from the measurement with a $^{60}$Co source (activity $A = 3.08(5) \times 10^5$ Bq) is indicated as well[30]. **d** In this graph, we show the prior and posterior predictive distributions using the 99% central credible interval obtained by the sum mode inversion pipeline. In addition, the experimental data $\mathbf{y}$ together with the derived posterior predictions using point estimators, i.e. the MAP probability estimate $\mathbf{x}_{MAP}$, the posterior mean $\mathbf{x}_{Mean}$ and the posterior median $\mathbf{x}_{Median}$, are indicated. **e** We show the total Sobol indices $S^T$ computed by the polynomial chaos expansion emulator[49] as a function of the spectral energy for the individual model parameters on the sum channel. **f** This graph presents the total ($\sigma_{tot}$) and the intrinsic ($\sigma_{intr}$) spectral resolution for the adopted detector system characterized by the standard deviation $\sigma$ as a function of the photon energy $E_\gamma$. The empirical resolution data as well as the corresponding resolution model were presented already elsewhere[30]. For the zoomed inset with $E_\gamma < 110$ keV, the K-absorption edge for iodine K[I] is highlighted[55]. For all graphs presented in this figure, uncertainties are provided as 1 standard deviation (SD) values (coverage factor $k = 1$).

consistent results using the single-mode inversion pipeline (Supplementary Fig. S15). Furthermore, by comparing the posterior Compton edge predictions for the sum channel, we find no statistically significant difference between the two inversion pipelines (Supplementary Fig. S16). From a modeling perspective, it is interesting to add that we observe no significant difference for Compton edge predictions using the various point estimators discussed in the previous section for both inversion pipelines.

## Intrinsic resolution

As already mentioned in the introduction, the scintillation non-proportionality not only distorts the spectral features in the pulse-height spectra but deteriorates also the spectral resolution of a scintillator detector. This contribution to the overall resolution due to the scintillation non-proportionality will be referred to as intrinsic resolution $\sigma_{intr}$ in accordance with previous studies[25–28,50–52]. The intrinsic resolution is of great importance for two key reasons.

First, it sets a fundamental lower limit on the achievable spectral resolution for a given scintillator material, making it a crucial factor in the development of new scintillators. As an example, in 1991, the scintillator $Lu_2SiO_5$ (LSO) was developed as an alternative to other available options at that time, such as $Bi_4Ge_3O_{12}$ (BGO). However, the performance of LSO led to considerable confusion within the research community as LSO exhibits a light yield more than four times greater than that of BGO, yet their energy resolutions are comparable[16]. Consequently, the energy resolution for LSO was not dominated by counting statistics but some other factor. Thanks to the development of the Compton coincidence measurement technique in 1994[12], subsequent experimental studies have conclusively shown that the pronounced scintillation non-proportionality of the LSO scintillator was indeed the underlying factor responsible for the observed resolution degradation[28,53]. This example showcases the need for a better understanding and prediction of the intrinsic resolution in the development of new scintillators[54].

Second, from a more technical perspective, the intrinsic resolution is a key component in the postprocessing pipeline for Monte Carlo simulations including NPSMs (Methods). In the forward model discussed before, the transport of scintillation photons, signal amplification by the photomultiplier tube and subsequent signal postprocessing in the multichannel analyzer are not included. As a result, to account for the additional resolution degradation by these processes, we need to perform a spectral broadening operation using a dedicated energy resolution model based on the measured total energy resolution as well as the intrinsic contribution[1].

Since our forward model explicitly accounts for the non-proportional scintillation physics by adopting an NPSM, we can use this numerical tool not only to predict pulse-height spectra but also to characterize the intrinsic resolution. We adopted a set of multiple monoenergetic Monte Carlo simulations to quantify the intrinsic resolution for different photon energies (Methods). Using this dataset, we then trained a Gaussian process (GP) regression model to predict the intrinsic resolution characterized by the standard deviation $\sigma$ for a given photon energy $E_\gamma$. The resulting GP model predictions together with the intrinsic data are highlighted in Fig. 3f. In the same graph, we include also the empirical model to describe the overall energy resolution $\sigma_{tot}$ as well as the corresponding empirical dataset[30].

Comparing the intrinsic and overall spectral resolution, we find an almost constant ratio $\sigma^2_{intr}/\sigma^2_{tot} \approx 0.35$ for $E_\gamma \gtrsim 1500$ keV. Around $E_\gamma \approx 440$ keV, there is a pronounced peak with $\sigma^2_{intr}/\sigma^2_{tot} \approx 0.42$ and for $E_\gamma \lesssim 440$ keV, we observe a significant decrease in $\sigma^2_{intr}/\sigma^2_{tot}$ with decreasing photon energy $E_\gamma$. Moreover, we find a more complex behavior in $\sigma_{intr}$ for $E_\gamma \lesssim 110$ keV. For $28$ keV $\lesssim E_\gamma \lesssim 60$ keV, the K-absorption edge for iodine K[I] at $E_\gamma = 33.1694(4)$ keV[55] alters the resolution significantly. On the other hand, at even smaller photon energies, there is again a pronounced increase in $\sigma_{intr}$ with decreasing energy compared to the mere moderate increase for $60$ keV $\lesssim E_\gamma \lesssim 110$ keV.

### Bayesian calibrated NPSM simulations

In addition to the insights into the Compton edge dynamics as well as the intrinsic resolution, the Bayesian inferred NPSM in combination with our forward model offers also the possibility to predict the full spectral detector response for new radiation sources accounting for non-proportional scintillation effects over the entire spectral range of our detector system. We used the $^{88}Y$ ($A = 6.83(14) \times 10^5$ Bq) and $^{137}Cs$ ($A = 2.266(34) \times 10^5$ Bq) radiation measurements to validate our calibrated NPSM. For the Monte Carlo simulations, we applied the posterior point estimators $x_{MAP}$ obtained by the sum mode inversion pipeline in combination with the intrinsic and total resolution models discussed in the previous sections.

In Fig. 4, we present the measured and simulated spectral detector response for $^{88}Y$ and $^{137}Cs$ together with $^{60}Co$, whose Compton edge

domain was used to perform the Bayesian inversion. For the simulations, we adopted a standard proportional scintillation model as well as the Bayesian inferred NPSM presented in this study. In line with the Compton scattering theory (Supplementary Methods S1.4), we find an enhanced shift of the Compton edge toward smaller spectral energies as the photon energy increases. For all three measurements, we observe a significant improvement in the Compton edge prediction for the NPSM simulations compared to the standard proportional approach. However, there are still some discrepancies at the lower end of the Compton edge domain. Moreover, we find also some deviations between the Compton edge and the FEP for $^{88}Y$ and $^{137}Cs$. It is important to note that these discrepancies are smaller or at least of similar size for the NPSM simulations compared to the proportional approach indicating that the former performs statistically significantly better over the entire spectral domain. Additional validation results for $^{57}Co$, $^{109}Cd$, $^{133}Ba$, and $^{152}Eu$ together with a detailed uncertainty analysis for each source are attached in the Supplementary Information File for this study (Supplementary Figs. S18–S25).

## Discussion

Here, we demonstrated that Compton edge probing combined with Monte Carlo simulations and Bayesian inversion can successfully infer NPSMs for NaI(Tl) inorganic scintillators. A detailed Bayesian data analysis revealed no significant differences between standard posterior point estimators and the related spectral detector response predictions for both inversion pipelines. Consequently, the Bayesian inversion results indicate that our methodology successfully constrained the NPSM parameters to a unique solution. However, we found statistically significant differences between our results and best-estimate literature values as well as between the individual scintillation crystals themselves. These results corroborate the experimental findings of Hull and his co-workers[34] and underscore the criticality of the NPSM calibration for every individual detector system.

Various studies reported a distortion of the Compton edge in gamma-ray spectrometry with inorganic scintillators[1,14,15,29,30]. In this study, we presented conclusive evidence that this shift is, at least partly, the result of the scintillation non-proportionality. Moreover, using our numerical models, we can predict the Compton edge shift as a function of the NPSM parameters. We observed a Compton edge shift toward smaller spectral energies for an increase in $dE/ds|_{Birks}$ and $\eta_{e/h}$ as well as a decrease in $dE/ds|_{Trap}$. These results imply that an enhanced scintillation non-proportionality promotes a Compton edge shift toward smaller spectral energies. In line with these observations, the non-proportionality is enhanced by a large $e^-/h$ fraction, an increased Birks mechanism as well as a reduction in the $e^-/h$ trapping rate[20,24,46].

Further, we quantified the sensitivity of the NPSM on the individual NPSM parameters using a PCE-based Sobol decomposition approach. The sensitivity results indicate that $\eta_{e/h}$ has the highest sensitivity on the Compton edge, followed by $dE/ds|_{Birks}$ and $dE/ds|_{Trap}$. However, previous studies showed a pronounced dependence of $dE/ds|_{Trap}$ on the ambient temperature[22,33]. In addition, we expect also a substantial change of the crystal structure by radiation damage, i.e. the creation of point defects in harsh radiation environments[10,56]. Therefore, the obtained sensitivity results should be interpreted with care. $dE/ds|_{Trap}$ might be of significant importance to model the dynamics in the detector response with changing temperature or increase in radiation damage to the crystals, e.g. in deep space missions.

Using the Bayesian calibrated NPSM, we are also able to numerically characterize the contribution of the scintillation non-proportionality to the overall energy resolution. This intrinsic resolution sets a fundamental lower limit on the achievable spectral resolution for a given scintillator material, making it a key factor in the development of new scintillators. At higher photon energies

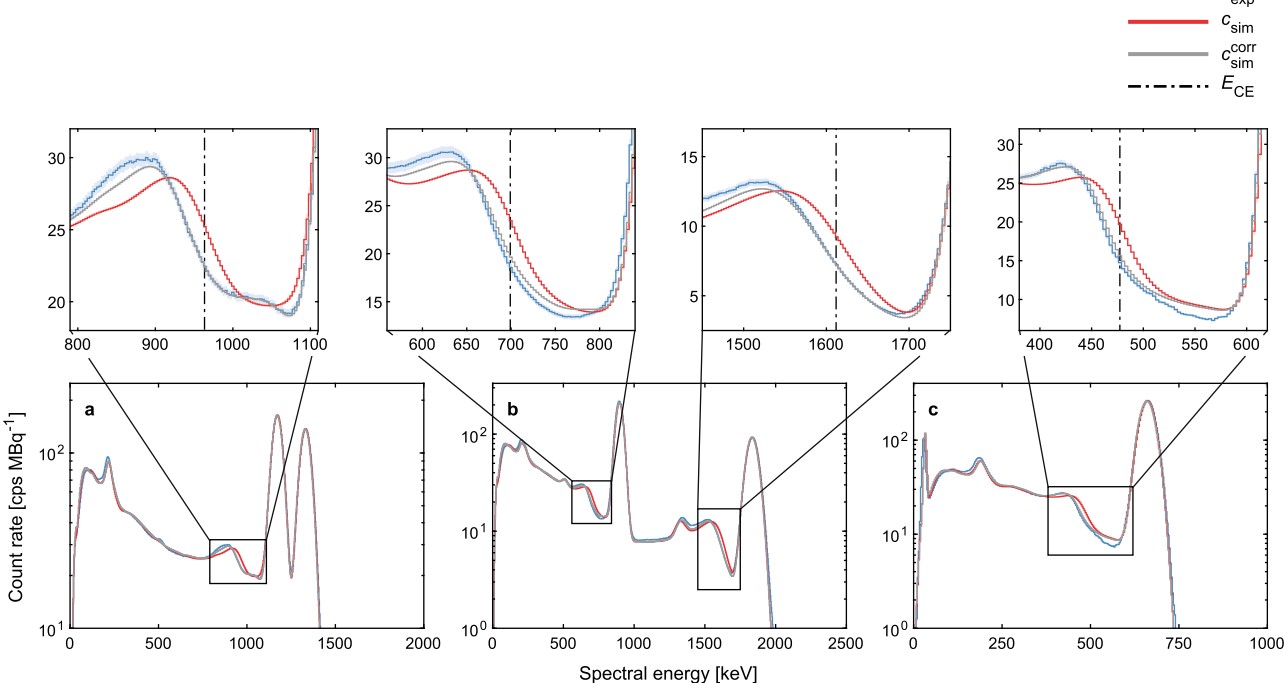

**Fig. 4 | Simulated spectral detector response using a Bayesian calibrated non-proportional model.** The measured and simulated spectral detector responses are shown for three different calibrated radionuclide sources: **a** $^{60}$Co ($A = 3.08(5) \times 10^5$ Bq). **b** $^{88}$Y ($A = 6.83(14) \times 10^5$ Bq). **c** $^{137}$Cs ($A = 2.266(34) \times 10^5$ Bq). The zoomed-in subfigures highlight the Compton edge region and include also the Compton edge $E_{CE}$ predicted by the Compton scattering theory[10], i.e. [477.334(3), 699.133(3), 963.419(3), 1611.77(1)] keV associated with the photon emission lines at [661.657(3), 898.042(3), 1173.228(3), 1836.063(3)] keV, respectively. The measured net count rate $c_{exp}$ as well as the simulated net count rate adopting a proportional scintillation model $c_{sim}$ were presented already elsewhere[30]. We obtained the simulated net count rate $c_{sim}^{corr}$ the same way as $c_{sim}$ but accounted for the non-proportional scintillation effects by the Bayesian calibrated NPSM obtained by the sum mode inversion pipeline. For the calibration, we used the $^{60}$Co dataset. For all graphs presented in this figure, uncertainties are provided as 1 standard deviation (SD) shaded areas (coverage factor $k = 1$). These uncertainties are only visible for $c_{exp}$.

($E_\gamma \gtrsim 400$ keV), we observed a significant contribution to the total spectral resolution ($\gtrsim$35%) with a maximum of $\approx$ 42% around 440 keV. At lower energies (10 keV $\lesssim E_\gamma \lesssim$ 400 keV), the intrinsic contribution is reduced and shows substantial distortions around the K-absorption edge for iodine at about 33 keV. We conclude that the non-proportional scintillation is a significant contributor to the total energy resolution of NaI(Tl). These observations are in good agreement with previous results[28,52,57–60] and thereby substantiate not only the predictive power of our numerical model but showcase also its potential as a valuable tool in the development of new scintillators.

Most of the theoretical studies focused on the prediction of NPSMs themselves. In contrast, available numerical models to predict the full detector response are scarce, computational intense and complex due to the adopted multi-step approaches with offline convolution computations[52,53,58]. In this study, we present an alternative way to implement NPSMs and simulate the full spectral detector response to gamma-ray fields by directly evaluating the NPSM online during the Monte Carlo simulations. This approach saves considerable computation time and has the additional advantage of not having to store and analyze large files with secondary particle data. We have used this implementation to predict the full spectral detector response for additional radiation fields accounting for non-proportional scintillation effects. Validation measurements revealed a significant improvement in the simulated detector response compared to proportional scintillation models. However, there are still some model discrepancies, especially at the lower and higher end of the Compton edge domain. These discrepancies might be attributed to systematic uncertainties in the Monte Carlo mass model or deficiencies in the

adopted NPSM. Sensitivity analysis performed in a previous study in conjunction with the prior prediction density results might indicate the latter[30].

While we focused our work on NaI(Tl) in electron and gamma-ray fields, the presented methodology can easily be extended to a much broader range of applications. First, it is general consensus that the light yield as a function of the stopping power is, at least to a first approximation, independent of the ionizing particle type[16,31]. Second, the adopted NPSM was validated with an extensive database of measured scintillation light yields for inorganic scintillators, i.e. BGO, CaF$_2$(Eu), CeBr$_3$, Cs(Tl), Cs(Na), LaBr$_3$(Ce), LSO(Ce), NaI(Tl), SrI$_2$, SrI$_2$(Eu), YAP(Ce) and YAG(Ce), among others[17,18,22]. From this, it follows that, given a gamma-ray field with resolvable Compton edges can be provided, our methodology may in principle be applied to any combination of inorganic scintillator and ionizing radiation field, including protons, $\alpha$-particles and heavy ions. However, it is important to note that our methodology relies on the observation of Compton edge shifts with a sufficient signal-to-noise ratio (SNR). We have shown that these shifts are influenced by the strength of scintillation non-proportionality of a given scintillator. As a result, scintillator materials that exhibit only a mild non-proportional scintillation response, e.g. LaBr$_3$(Ce) or YAP(Ce), may present challenges for the calibration of an NPSM due to the reduced SNR in the Compton edge shift. Further investigations are required to assess the applicability of our methodology to such scintillators. That said, the presented methodology can be readily adapted using Bayes's theorem to address low SNR cases more effectively by combining multiple Compton edge domains or by probing additional spectral features distorted by the non-proportional scintillation response.

In summary, we conclude that NPSMs are essential for accurate detector response simulations, especially for scintillators with large crystal volumes[25,28,31], e.g. in dark matter research, total absorption spectroscopy or remote sensing[1–7,30]. The methodology presented in this study offers a reliable and cost-effective alternative to existing experimental methods to investigate non-proportional scintillation physics phenomena and perform accurate full detector response predictions with Bayesian calibrated NPSM. Moreover, it does not require any additional measurement equipment and can therefore be applied for any inorganic scintillator spectrometer, also during detector deployment. This is especially attractive for applications, where the scintillator properties change in operation, e.g. due to radiation damage or temperature changes, but also for detector design and the development of advanced scintillator materials. Last but not least, we can use the derived numerical models not only for NPSM inference but also to investigate and predict various scintillator properties, e.g. intrinsic resolution or Compton edge dynamics, and thereby contribute to a better understanding of the complex scintillation physics in inorganic scintillators.

## Methods

### Gamma-ray spectrometry

We performed gamma-ray spectrometric measurements in the calibration laboratory at the Paul Scherrer Institute (PSI) (inner room dimensions: 5.3 m × 4.5 m × 3 m). The adopted spectrometer consisted of four individual 10.2 cm × 10.2 cm × 40.6 cm prismatic NaI(Tl) scintillation crystals with the associated photomultiplier tubes and the electronic components embedded in a thermal-insulating and vibration-damping polyethylene foam protected by a rugged aluminum detector box (outer dimensions: 86 cm × 60 cm × 30 cm). The spectrometer features 1024 channels for an energy range between about 30 and 3000 keV together with automatic linearization of the individual scintillation crystal spectra[30]. We used seven different calibrated radionuclide sources ($^{57}$Co, $^{60}$Co, $^{88}$Y, $^{109}$Cd, $^{133}$Ba, $^{137}$Cs, and $^{152}$Eu) from the Eckert & Ziegler Nuclitec GmbH. We inserted these sources consisting of a radionuclide carrying ion exchange sphere (diameter 1 mm) embedded in a 25 mm × 3 mm solid plastic disc into a custom low absorption source holder made out of a polylactide polymer (PLA) and placed this holder on a tripod in a fixed distance of 1 m to the detector front on the central detector $x$-axis. To measure the source-detector distances and to position the sources accurately, distance as well as positioning laser systems were used. A schematic depiction of the measurement setup is shown in Fig. 1a.

Between radiation measurements, background measurements were performed regularly for background correction and gain stability checks. For all measurements, the air temperature as well as the air humidity in the calibration laboratory was controlled by an air conditioning unit and logged by an external sensor. The air temperature was set at 18.8(4) °C and the relative air humidity at 42(3)%. The ambient air pressure, which was also logged by the external sensor, fluctuated around 982(5) hPa.

During measurements, additional instruments and laboratory equipment were located in the calibration laboratory, e.g. shelves, a workbench, a source scanner or a boiler as shown in Fig. 1a. The effect of these features on the detector response was carefully assessed in ref. 30.

After postprocessing the spectral raw data according to the data reduction pipelines described in ref. 30, we extracted the Compton edge spectral data from the net count rate spectra. The spectral domain of the Compton edge $\mathcal{D}_E$ was defined as $\mathcal{D}_E := \{E : E_{CE} - 3 \cdot \sigma_{tot}(E_{CE}) \leq E \leq E_{FEP} - 2 \cdot \sigma_{tot}(E_{FEP})\}$, where $E$ is the spectral energy, $\sigma_{tot}$ the energy dependent total resolution characterized by the standard deviation[30] and $E_{FEP}$ the FEP associated with the Compton edge $E_{CE}$. Neglecting Doppler broadening and atomic shell effects, we compute $E_{CE}$ according to the Compton scattering theory[10] as follows:

$$E_{CE} := E_{FEP}\left(1 - \frac{1}{1 + \frac{2E_{FEP}}{m_e c^2}}\right) \tag{2}$$

where $m_e c^2$ is defined as the energy equivalent electron mass. In this study, we consulted the ENDF/B-VIII.0 nuclear data file library[61] for nuclear decay-related data as well as the Particle Data Group library[62] for fundamental particle properties.

To investigate the sensitivity of the selected Compton edge domain $\mathcal{D}_E$ on the Bayesian inversion results, we performed a sensitivity analysis on $\mathcal{D}_E$. Within the uncertainty bounds, the inversion results have proven to be insensitive to small alterations in $\mathcal{D}_E$ (Supplementary Table S3).

It is important to note that, if not otherwise stated, uncertainties are provided as 1 standard deviation (SD) values in this study (coverage factor $k = 1$). For more information about the radiation measurements and adopted data reduction pipelines, the reader is referred to the dedicated study[30] as well as the Supplementary Information File for this work (Supplementary Methods S1.3).

### Monte Carlo simulations

We performed all simulations with the multi-purpose Monte Carlo code FLUKA version 4.2.1[44] together with the graphical interface FLAIR version 3.1-15.1[63]. We used the most accurate physics settings (precisio) featuring a high-fidelity fully coupled photon, electron and positron radiation transport for our source-detector configuration. In addition, this module accounts for secondary electron production and transport, Landau fluctuations as well as X-ray fluorescence, all of which are essential for an accurate description of non-proportional scintillation effects[16,18,23,52]. Motivated by the range of the transported particles, lower kinetic energy transport thresholds were set to 1 keV for the scintillation crystals as well as the closest objects to the crystals, e.g. reflector, optical window and aluminum casing for the crystals. For the remaining model parts, the transport threshold was set to 10 keV to decrease the computational load while maintaining the high-fidelity transport simulation in the scintillation crystals. All simulations were performed on a local computer cluster (7 nodes with a total number of 520 cores at a nominal clock speed of 2.6 GHz) at the Paul Scherrer Institute utilizing parallel computing.

We scored the energy deposition events in the scintillation crystals individually on an event-by-event basis using the custom user routine usreou together with the detect card. The number of primaries was set to $10^7$ for all simulations, which guarantees a maximum relative statistical standard deviation $\sigma_{stat,sim,k}/c_{sim,k} < 1\%$ and a maximum relative variance of the sample variance $VOV_k < 0.01\%$ for all detector channels $k$. More details on the simulation settings as well as on the postprocessing of the energy deposition data can be found in ref. 30.

To implement the NPSM described by Eq. (1), we developed an additional user routine comscw. Similar to refs. 1,64, we weight each individual energy deposition event in the scintillator, point-like or along the charged particle track, by the scintillation light yield given in Eq. (1) (Supplementary Algorithm S1). The resulting simulated response is then rescaled to match the energy calibration models derived in ref. 30. Using our methodology, we get simulated pulse-height spectra that incorporate non-proportional effects across the entire spectral range of our detector system.

### Surrogate modeling

We applied a custom machine learning trained vector-valued polynomial chaos expansion (PCE) surrogate model to emulate the spectral Compton edge detector response over $\mathcal{D}_E$ for both, the individual scintillation crystals as well as the sum channel. PCE models are ideal

candidates to emulate expensive-to-evaluate vector-valued computational models[43]. As shown in refs. [65–67], any function $\mathbf{Y} = \mathcal{M}(\mathbf{X})$ with the random input vector $\mathbf{X} \in \mathbb{R}^{M \times 1}$ and random response vector $\mathbf{Y} \in \mathbb{R}^{N \times 1}$ can be expanded as a so-called polynomial chaos expansion provided that $\mathbb{E}[|\mathbf{Y}|^2] < \infty$:

$$\mathbf{Y} = \mathcal{M}(\mathbf{X}) = \sum_{\boldsymbol{\alpha} \in \mathbb{N}^M} \mathbf{a}_{\boldsymbol{\alpha}} \Psi_{\boldsymbol{\alpha}}(\mathbf{X}) \tag{3}$$

where $\mathbf{a}_{\boldsymbol{\alpha}} := (a_{1,\boldsymbol{\alpha}}, \ldots, a_{N,\boldsymbol{\alpha}})^\top \in \mathbb{R}^{N \times 1}$ are the deterministic expansion coefficients, $\boldsymbol{\alpha} := (\alpha_1, \ldots, \alpha_M)^\top \in \mathbb{N}^{M \times 1}$ the multi-indices storing the degrees of the univariate polynomials $\psi_\alpha$ and $\Psi_{\boldsymbol{\alpha}}(\mathbf{X}) := \prod_{i=1}^M \psi_{\alpha_i}^i(X_i)$ the multivariate polynomial basis functions, which are orthonormal with respect to the joint probability density function $f_{\mathbf{X}}$ of $\mathbf{X}$, i.e. $\langle \Psi_{\boldsymbol{\alpha}}, \Psi_{\boldsymbol{\beta}} \rangle_{f_{\mathbf{X}}} = \delta_{\boldsymbol{\alpha},\boldsymbol{\beta}}$.

To reduce the computational burden, we combined the PCE model with principal component analysis (PCA) allowing us to characterize the main spectral Compton edge features of the response by means of a small number $N'$ of output variables compared to the original number $N$ of spectral variables, i.e. $N' \ll N$[43]. Similar to ref. [68], we computed the emulated computational model response $\hat{\mathcal{M}}_{\text{PCE}}(\mathbf{X})$ in matrix form as

$$\mathbf{Y} \approx \hat{\mathcal{M}}_{\text{PCE}}(\mathbf{X}) = \boldsymbol{\mu}_{\mathbf{Y}} + \text{diag}(\boldsymbol{\sigma}_{\mathbf{Y}}) \boldsymbol{\Phi}' \mathbf{A} \boldsymbol{\Psi}(\mathbf{X}) \tag{4}$$

with $\boldsymbol{\mu}_{\mathbf{Y}}$ and $\boldsymbol{\sigma}_{\mathbf{Y}}$ being the mean and standard deviation of the random vector $\mathbf{Y}$ and $\boldsymbol{\Phi}'$ the matrix containing the retained eigenvectors $\boldsymbol{\phi}$ from the PCA, i.e. $\boldsymbol{\Phi}' := (\boldsymbol{\phi}_1, \ldots, \boldsymbol{\phi}_{N'}) \in \mathbb{R}^{N \times N'}$. On the other hand, the vector $\boldsymbol{\Psi}(\mathbf{X}) \in \mathbb{R}^{\text{card}(\mathcal{A}^\star) \times 1}$ and matrix $\mathbf{A} \in \mathbb{R}^{N' \times \text{card}(\mathcal{A}^\star)}$ store the multivariate orthonormal polynomials and corresponding PCE coefficients, respectively. The union set $\mathcal{A}^\star := \bigcup_{j=1}^{N'} \mathcal{A}_j$ includes the finite sets of multi indices $\mathcal{A}_j$ for the $N'$ output variables following a specific truncation scheme.

We used a Latin hypercube experimental design $\mathcal{X} \in \mathbb{R}^{M \times K}$[69,70] with $K = 200$ instances sampled from a probabilistic model, which itself is defined by the model parameter priors described in the next subsection. The model response $\mathcal{Y} \in \mathbb{R}^{N \times K}$ for this design was then evaluated using the forward model described in the previous subsection. We adopted a hyperbolic truncation scheme $\mathcal{A}_j := \{\boldsymbol{\alpha} \in \mathbb{N}^M : (\sum_{i=1}^M \alpha_i^q)^{1/q} \le p\}$ with $p$ and $q$ being hyperparameters defining the maximum degree for the associated polynomial and the q-norm, respectively. To compute the PCE coefficient matrix $\mathbf{A}$, we applied adaptive least angle regression[71] and optimized the hyperparameters $p := \{1, 2, \ldots, 7\}$ and $q := \{0.5, 0.6, \ldots, 1\}$ using machine learning with a holdout partition of 80% and 20% for the training and test set, respectively. For the PCA truncation, we adopted a relative PCA-induced error $\varepsilon_{\text{PCA}}$ of 0.1%, i.e. $N' := \min\{S \in \{1, \ldots, N\} : \sum_{j=1}^S \lambda_j / \sum_{j=1}^N \lambda_j \ge 1 - \varepsilon_{\text{PCA}}\}$ with $\lambda$ being the eigenvalues from the PCA. The resulting generalization error of the surrogate models, characterized by the relative mean squared error over the test sets, are <1% and <2% for the sum channel and the individual scintillation crystals, respectively. All PCE computations were performed with the `UQLab` code[72] in combination with custom scripts to perform the PCA. More information about the PCE-PCA models as well as the PCE-PCA-based Sobol indices including detailed derivations are included in the Supplementary Information File for this study (Supplementary Methods S1.1–S1.2).

## Bayesian inference
Following the Bayesian framework[40], we approximate the measured spectral detector response $\mathbf{y} \in \mathbb{R}^{N \times 1}$ with a probabilistic model combining the forward model $\mathcal{M}(\mathbf{x}_{\mathcal{M}})$ and model parameters $\mathbf{x}_{\mathcal{M}} \in \mathbb{R}^{M_{\mathcal{M}} \times 1}$ with an additive discrepancy term $\boldsymbol{\varepsilon}$, i.e. $\mathbf{y} := \mathcal{M}(\mathbf{x}_{\mathcal{M}}) + \boldsymbol{\varepsilon}$. For the discrepancy term $\boldsymbol{\varepsilon}$, which characterizes the measurement noise and prediction error, we assume a Gaussian model $\pi(\boldsymbol{\varepsilon}|\sigma_\varepsilon^2) = \mathcal{N}(\boldsymbol{\varepsilon}|\mathbf{0}, \sigma_\varepsilon^2 \mathbb{I}_N)$ with unknown discrepancy variance $\sigma_\varepsilon^2$. On the other hand, as

discussed in the previous subsection, we emulate the forward model $\mathcal{M}(\mathbf{x}_{\mathcal{M}})$ with a PCE surrogate model $\hat{\mathcal{M}}_{\text{PCE}}(\mathbf{x}_{\mathcal{M}})$. Consequently, we can compute the likelihood function as follows:

$$\pi(\mathbf{y}|\mathbf{x}) = \mathcal{N}\left(\mathbf{y}|\hat{\mathcal{M}}_{\text{PCE}}(\mathbf{x}_{\mathcal{M}}), \sigma_\varepsilon^2 \mathbb{I}_N\right) \tag{5}$$

with $\mathbf{x} := [\mathbf{x}_{\mathcal{M}}, \sigma_\varepsilon^2]^\top$ and $\mathbf{x}_{\mathcal{M}} := [dE/ds|_{\text{Birks}}, \eta_{e/h}, dE/ds|_{\text{Trap}}]^\top$. In combination with the prior density $\pi(\mathbf{x})$, we can then compute the posterior distribution using Bayes' theorem[42]:

$$\pi(\mathbf{x} \mid \mathbf{y}) = \frac{\pi(\mathbf{y} \mid \mathbf{x}) \pi(\mathbf{x})}{\int_{\mathcal{D}_{\mathbf{x}}} \pi(\mathbf{y} \mid \mathbf{x}) \pi(\mathbf{x}) \, d\mathbf{x}} \tag{6}$$

where we assume independent marginal priors, i.e. $\pi(\mathbf{x}) = \prod_{i=1}^M \pi(x_i)$ with $M = M_{\mathcal{M}} + 1$. We defined the marginal priors based on the principle of maximum entropy[73] as well as empirical data from previous studies[17,18,22]. It should be emphasized that we applied the sum mode inversion pipeline first followed by the single mode inversion pipeline. In accordance with Bayes' theorem[42], we therefore incorporate the results obtained by the sum mode inversion pipeline in the marginal priors used for the single mode inversion pipeline. A full list of all adopted marginal priors for both pipelines is attached in the Supplementary Information File for this study (Supplementary Table S1). Using the prior and posterior distributions, we can then also make predictions on future model response measurements $\mathbf{y}^*$ leveraging the prior and posterior predictive densities:

$$\pi(\mathbf{y}^*) = \int_{\mathcal{D}_{\mathbf{x}}} \pi(\mathbf{y}^*|\mathbf{x}) \pi(\mathbf{x}) \, d\mathbf{x} \tag{7a}$$

$$\pi(\mathbf{y}^*|\mathbf{y}) = \int_{\mathcal{D}_{\mathbf{x}}} \pi(\mathbf{y}^*|\mathbf{x}) \pi(\mathbf{x}|\mathbf{y}) \, d\mathbf{x} \tag{7b}$$

All Bayesian computations were performed with the `UQLab` code[72]. We applied an affine invariant ensemble algorithm[39] to perform Markov Chain Monte Carlo (MCMC) and thereby estimate the posterior distribution $\pi(\mathbf{x}|\mathbf{y})$. We used 10 parallel chains with $2 \times 10^4$ MCMC iterations per chain together with a 50% burn-in. The convergence and precision of the MCMC simulations were carefully assessed using standard diagnostics tools[42,74]. We report a potential scale reduction factor $\hat{R} < 1.02$ and an effective sample size ESS $\gg 400$ for all performed MCMC simulations. Additional trace and convergence plots for the individual parameters $\mathbf{x}$ and point estimators (Supplementary Figs. S1–S10), a full list of the Bayesian inversion results (Supplementary Table S2) as well as a sensitivity analysis on the adopted Compton edge domain (Supplementary Table S3) can be found in the attached Supplementary Information File for this study.

## Resolution modeling
In this last section, we will discuss the derivation of the energy resolution models adopted in this study. We start with the model to characterize the overall or total energy resolution $\sigma_{\text{tot}}$ for our detector system and describe in a second step the derivation of the intrinsic resolution model $\sigma_{\text{intr}}$. It is important to note that in contrast to $\sigma_{\text{intr}}$, we provide here only a short summary of the key aspects involved in $\sigma_{\text{tot}}$. The entire postprocessing pipeline to derive $\sigma_{\text{tot}}$ was already thoroughly discussed in a previous study[30]. For more details, we kindly refer the reader to the dedicated study.

For each scintillation crystal, we quantified $\sigma_{\text{tot}}$ by characterizing the spectral dispersion of measured FEPs associated with known photon emission lines from specific radionuclides. The corresponding pipeline can be divided into three steps. In the first step, we extracted specific spectral domains containing a singlet or multiplet of FEPs from a set of measured count rate spectra covering a spectral range between 122 and 1836 keV[75]. In a second step, we fitted a spectral peak model

based on a sum of independent Gaussian peaks together with a numerical baseline[76] to the selected singlets or multiplets using weighted non-linear least-squares (WNLLS) regression combined with the interior-reflective Newton method[77]. In the third step, we extracted the Gaussian standard deviation parameters from the fitted FEPs as a characteristic measure for the spectral resolution. By combining these empirical resolution values with the known emission line energies, we derived an exponential model to describe $\sigma_{tot}$ as a function of the photon energy $E_\gamma$ adopting again WNLLS. The resulting relative generalization error, characterized by leave-one-out cross-validation, is <0.2% for all scintillation crystals.

To derive a model for $\sigma_{intr}$, we performed in the first step additional Monte Carlo simulations for an isotropic and uniform mono-energetic photon flux of energy $10\,\text{keV} \leq E_\gamma \leq 3200\,\text{keV}$. To account for the different spectral scales, we applied a non-uniform experimental design for the photon energy $E_\gamma$ with a 2 keV spacing below 110 keV and 100 keV spacing above. Moreover, to account for the non-proportional scintillation physics, we ran all simulations with the Bayesian calibrated NPSM, i.e. the derived MAP point estimators. The mass model for those simulations features a $10.2\,\text{cm} \times 10.2\,\text{cm} \times 40.6\,\text{cm}$ prismatic NaI(Tl) scintillation crystal embedded in a vacuum environment. In the second step, we extracted the mean light yield values from the simulated FEPs (Supplementary Algorithm S1). Similar to the measured spectra, we can then derive a simple polynomial energy calibration model using WNLLS to convert the simulated light yield to energy[30]. In a third step, we adopted the extracted $\sigma_{intr}$ from the individual energy-calibrated FEPs to train a Gaussian Process (GP) regression model with[78]:

$$\sigma_{intr}\left(E_\gamma\right) \sim \mathcal{GP}\left(\mathbf{f}\left(E_\gamma\right)^\top \boldsymbol{\beta}, \kappa\left(E_\gamma, E_\gamma'\right) + \sigma_{\mathcal{GP}}^2 \delta_{E_\gamma, E_\gamma'}\right) \quad (8)$$

where we applied a polynomial trend function of the second order, i.e. $\mathbf{f}(E_\gamma) := (1, E_\gamma, E_\gamma^2)^\top$ and $\boldsymbol{\beta} := (\beta_0, \beta_1, \beta_2)^\top$, a homoscedastic noise model with the noise variance $\sigma_{\mathcal{GP}}^2$ and Kronecker delta $\delta_{E_\gamma, E_\gamma'}$ as well as a Matérn-3/2 covariance function $\kappa(E_\gamma, E_\gamma') := (1 + \sqrt{3}|E_\gamma - E_\gamma'|/\theta) \exp\left(-\sqrt{3}|E_\gamma - E_\gamma'|/\theta\right)$ with the kernel scale $\theta$. It is worth noting that, due to the known asymmetry in the FEPs[1,25,28], we adopted numerical estimates both for the mean and standard deviation parameters associated with the individual FEPs. With the $N$-dimensional intrinsic dataset $\{\mathbf{E}_\gamma, \boldsymbol{\sigma}_{intr}\}$, we can then predict the intrinsic resolution $\boldsymbol{\sigma}_{intr}^*$ for a new set of $N^*$ photon energies $\mathbf{E}_\gamma^*$ using the GP posterior predictive density as follows[78]:

$$\pi\left(\boldsymbol{\sigma}_{intr}^* \mid \mathbf{E}_\gamma^*, \mathbf{E}_\gamma, \boldsymbol{\sigma}_{intr}\right) = \mathcal{N}\left(\boldsymbol{\sigma}_{intr}^* \mid \boldsymbol{\mu}_{\mathcal{GP}}, \boldsymbol{\Sigma}_{\mathcal{GP}}\right) \quad (9a)$$

$$\boldsymbol{\mu}_{\mathcal{GP}} = \mathbf{F}_*^\top \hat{\boldsymbol{\beta}} + \mathbf{K}_*^\top \mathbf{K}^{-1}\left(\boldsymbol{\sigma}_{intr} - \mathbf{F}^\top \hat{\boldsymbol{\beta}}\right) \quad (9b)$$

$$\boldsymbol{\Sigma}_{\mathcal{GP}} = \mathbf{K}_{**} - \mathbf{K}_*^\top \mathbf{K}^{-1} \mathbf{K}_* + \mathbf{U}^\top\left(\mathbf{F}\mathbf{K}^{-1}\mathbf{F}^\top\right)^{-1}\mathbf{U} \quad (9c)$$

$$\hat{\boldsymbol{\beta}} = \left(\mathbf{F}\mathbf{K}^{-1}\mathbf{F}^\top\right)^{-1}\mathbf{F}\mathbf{K}^{-1}\boldsymbol{\sigma}_{intr} \quad (9d)$$

$$\mathbf{U} = \mathbf{F}_* - \mathbf{F}\mathbf{K}^{-1}\mathbf{K}_* \quad (9e)$$

with the matrices $\mathbf{F} = \mathbf{f}(\mathbf{E}_\gamma) \in \mathbb{R}^{3 \times N}$, $\mathbf{F}_* = \mathbf{f}(\mathbf{E}_\gamma^*) \in \mathbb{R}^{3 \times N^*}$, $\mathbf{K} = \kappa(\mathbf{E}_\gamma, \mathbf{E}_\gamma) + \sigma_{\mathcal{GP}}^2 \mathbb{I}_N \in \mathbb{R}^{N \times N}$, $\mathbf{K}_* = \kappa(\mathbf{E}_\gamma, \mathbf{E}_\gamma^*) \in \mathbb{R}^{N \times N^*}$ and $\mathbf{K}_{**} = \kappa(\mathbf{E}_\gamma^*, \mathbf{E}_\gamma^*) \in \mathbb{R}^{N^* \times N^*}$. To account for the different spectral scales, we trained two GP models, one for $10\,\text{keV} \leq E_\gamma \leq 90\,\text{keV}$ and the other one for $90\,\text{keV} \leq E_\gamma \leq 3200\,\text{keV}$, using the MATLAB® code. For both models, we applied 5-fold cross-validation in combination with Bayesian optimization to determine the GP hyperparameters $\sigma_{\mathcal{GP}}^2$ and $\theta$. It is important to add that in case of

the experimental design $\mathcal{X}$ adopted to train the surrogate model, we ran the pipeline for $\sigma_{intr}$ with the corresponding set of NPSM parameters defined by $\mathcal{X}$.

As discussed already in the Results section, the intrinsic resolution is also a key component in the postprocessing pipeline for Monte Carlo simulations including NPSMs. Because the Monte Carlo simulations performed for the forward model only inherently include the intrinsic resolution, we need to perform a spectral broadening operation to account for the additional energy resolution degradation due to the transport of scintillation photons, signal amplification by the photo-multiplier tube and subsequent signal postprocessing in the multi-channel analyzer. Similar to ref. 1, we assume statistical independence between the resolution degradation due to the scintillation non-proportionality and the aforementioned neglected processes in the Monte Carlo simulations. We can then perform the broadening operation as described in ref. 30 with an adapted dispersion $\sqrt{\sigma_{tot}^2 - \sigma_{intr}^2}$. For completeness, we included this adapted dispersion model in Fig. 3f.

## Data availability

The radiation measurement raw data presented herein have been deposited on the ETH Research Collection repository under accession code https://doi.org/10.3929/ethz-b-000528920[75].

## Code availability

The FLUKA code[44] used for Monte Carlo radiation transport and detector response simulations is available at https://fluka.cern/. We adopted the graphical user interphase FLAIR[63], available at https://flair.web.cern.ch/flair/, to setup the FLUKA input files and create the mass model figures. The custom FLUKA user routines adopted in the Monte Carlo simulations have been deposited on the ETH Research Collection repository under accession code https://doi.org/10.3929/ethz-b-000595727[79]. Data processing, machine learning computation and figure creation was performed by the MATLAB® code in combination with the open-source toolbox UQLab[72] available at https://www.uqlab.com/.

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

## Acknowledgements

The authors gratefully acknowledge the technical support by Dominik Werthmüller for the execution of the Monte Carlo simulations on the computer cluster at the Paul Scherrer Institute. We also thank Eduardo Gardenali Yukihara for helpful discussions and advices. Further, we would like to express our gratitude to the Swiss Armed Forces and the National Emergency Operations Centre (NEOC) for providing the detector system. This research has been supported in part by the Swiss Federal Nuclear Safety Inspectorate (grant no. CTR00491 to all authors except F.C.).

## Author contributions

D.B. designed the study, supervised the project, performed the measurements, simulations, data postprocessing and wrote the manuscript. F.C. significantly contributed to the implementation of the NPSM in FLUKA. G.B. supervised the project. S.M. acquired the project funding. All authors contributed to the completion of the manuscript (D.B., F.C., G.B., M.K., and S.M.).

## Competing interests

The authors declare no competing interests.
