## [Peer Review File · Nature Communications]

Emulator-based Bayesian Inference on Non-Proportional Scintillation Models by Compton-Edge ProbingREVIEWER COMMENTS

Reviewer #1 (Remarks to the Author):

Scintillators are used in countless radiation measurement applications, in medicine, industry, exploration, and basic research. For gamma ray spectroscopy, an understanding of the nonproportional relationship between energy deposited and light output is needed for analysis of the data. Nonproportionality is also a limiting factor in achieving high energy resolution with scintillators. A method to rapidly assess the performance of new scintillators would be an important tool for developers.

The authors demonstrate the use of Compton-edge spectroscopy to determine parameters of the Birks-Onsager light yield model reported by Payne et al. (2014). They argue that their approach is more accessible to researchers than existing methods (“available in a limited number of laboratories”) and can be used to calibrate instruments for a myriad of applications. Their experimental setup is quite simple, consistent with that widely used for benchtop testing in industry. The physical principles and information constrained by Compton edge probing are presented in sufficient detail.

The analytical method is straightforward. Markov Chain Monte Carlo (MCMC) is widely used for parameter inversions via Bayesian inference. Prior to reading their manuscript, I was unfamiliar with Polynomial Chaos Expansion (PCE). However, the Methods provides an adequate description of the approach. Fast calculation of the forward model is required by for the MCMC inversion. This is where PCI comes in. The authors use the radiation transport code FLUKA to perform detailed calculations of the sensor response. The response is parameterized using PCE, resulting in a surrogate model that is used in the MCMC inversion. Depending on the computational resources available and approach to implementing the radiation transport model, the machine learning step might be bypassed in a future study. The surrogate model appears to have ample accuracy for the application.

Results are presented for sodium iodide – NaI(Tl), which is a widely used scintillator with significant nonproportionality. The inversion robustly estimated light output model parameters, producing results qualitatively like that reported by Payne et al. (2014) for NaI(Tl). Payne et al reported results for a wide temperature range. Nevertheless, it might be instructive to include for comparison temperature-interpolated values with the results of the MCMC inversion.

Limiting the current study to NaI(Tl) has the advantage of focusing on an exposition of the methodology. However, it would be helpful to see how well the methods work for other scintillators. The current generation of scintillators, including lanthanum bromide, have low nonproportionality. It is not clear how sensitive the Compton-edge probing method would be to light output parameters in this case. In addition, shifts in the gamma-ray photopeak positions also provide clues to nonproportionality. In

focusing exclusively on the edges, the authors are not using all the information available to them to characterize light yield.

In summary, the study provides a practical approach to determining light yield parameters, which would be of value to researchers using gamma ray spectroscopy and developing/assessing new materials.

Reviewer #2 (Remarks to the Author):

Reviewer's report

Evidences reported in the literature show that the light yield of inorganic scintillators is not proportional to the energy deposited in them by incident ionizing radiation (the authors provide numerous references on this subject). This is a real difficulty faced by one attempting to replicate with high fidelity the response, i.e., the measured spectrum, of such radiation detectors from simulations. To this end, the authors implemented a relation relating the light yield to the energy deposited (provided in equation (1) and based on findings in the literature) in their simulation model of a NaI(Tl) detector.

According to the literature, this relation contains parameters that depend on the properties of the given scintillator as well as on the environmental conditions. This means that the numerical value of those parameters are detector specific and subject to change (i.e., not universal and not fixed in time). The evaluation of this relation and of those parameters from optical and electron measurements can be laborious. Therefore, in this work, the authors propose, and demonstrate how, to evaluate efficiently those parameters via the well-established Bayesian inversion method applied to the Compton edge region.

From spectra obtained with various radioactive isotopes, the authors (1) demonstrate that the implementation of this non-proportional scintillation model in their simulation model does improve the agreement between the simulated and experimentally measured spectra, and (2) show evidences suggesting that this non-proportionality is partially responsible for the observed shift in energy in the Compton edge region (when comparing experimental measurements to the Compton edge theory).

The authors recognize that the non-proportionality corrected simulated spectra are still not entirely in agreements with the measured ones (at the edges of the Compton edge in particular). Nonetheless, the improvements are substantial and the scientific community would benefit from this work.

Moreover, to accelerate the Bayesian inversion, instead of using their simulation model directly, the authors proposed and developed an emulator, based on machine learning trained polynomial chaos expansion, able to determine the simulation results, in the Compton edge region, from given non-proportionality parameters values. This is a clever idea.

Throughout the manuscript and supplementary documentation, the authors provide pertinent information and justifications revealing the robustness and quality of their approach, methodology and analysis. For examples, (1) the description of the physical processes occurring from the radiation-matter interaction to the scintillation light production exhibits the limits of the non-proportional scintillation

model implemented, (2) the emulator is accompanied by its main equations, and its behavior is depicted within its range of validity (Fig. 3 a, b, c, and e), (3) the Bayesian inversion is accompanied by its main equations, by the convergence of the Markov Chain Monte Carlo algorithm (Fig. S1), by its priors distributions (Table 1), as well as by the convergence (Fig. S2), distributions (Fig. 2), and point estimators (Table S1) of the posteriors, (4) the simulated and experimentally measured data are accompanied by their uncertainties, and (5) the capability of the simulation model (with and without the non-proportional scintillation model) to reproduce the detector response is evaluated over a wide energy range using various radioactive isotopes (Figs. 3 d, 4, S3, S4, S5, S6, S7, S8, S9, and S10).

The manuscript and supplementary documentation contain limited information about the simulation model itself. For example, more details on the implementation in FLUKA would be appreciated. However, it is understood that this part of the simulation model was detailed in a previous publication, and that this manuscript focuses on new implementations.

Finally, the text is very well written, the figures are neat and the numerous references support adequately this work. Once the authors address the questions/concerns below and adapt the manuscript and/or the supplementary documentation accordingly, it will offer all the necessary information to understand the extent of the work for someone in the field of simulation of radiation detectors.

Questions to the authors

Question 1

Starting on line 26, you wrote: "Second, various studies stated the conjecture that specific spectral features such as the Compton edges are shifted and distorted as a result of the non-proportional scintillation response [1, 14, 15, 29, 30]." Why is this affecting the Compton edges more than other parts of the spectrum? Please explain.

Question 2

Starting on line 48, you wrote: "In this study, we propose Compton edge probing together with Bayesian inversion to infer and calibrate NPSMs." If the NPSM is calibrated using data taken from the Compton edge region, is it viable only in this region? In other words, in the forward simulation model, is the NPSM systematically applied to all energies in the spectrum, or limited to energies falling in Compton edge regions? Please clarify.

Question 3

Starting on line 67, you wrote: "The adopted spectrometer consisted of four individual 10.2 cm × 10.2 cm × 40.6 cm prismatic NaI(Tl) scintillation crystals." From this sentence and from Fig. 1a, I understand that this detector contains actually four crystals. Was the NPSM calibrated for each crystal individually, or after combining the crystals together? If the latter, as each crystal can be different, even just slightly, how is it affecting the calibrated NPSM? Please clarify.

Question 4

The plots of $dE/ds|_{\text{Trap}}$ at the bottom of Fig. 2 and in Fig. S1 c, show long tails and wide variations, respectively, all ranging from 10 to 15 MeV cm⁻¹. Would that be due to too wide priors, to an omitted physical process in the NPSM, or to the trapping process not being fully represented in the Compton edge region, so the Bayesian inversion method can't quantify that parameter with the same level of precision as the two other parameters? What are your thoughts about this?

Question 5

For example, starting on line 111, you wrote: "In addition, we fixed the Onsager related stopping power parameter $dE/ds|_{\text{Ons}}$ to 36.4 MeVcm⁻¹ as suggested by previous investigators [18, 22].", and you provided the value of the other parameters in Table S1. How would you explain that the parameters in equation (1) are all constant in energy (i.e., not energy dependent)?

Question 6

Starting on line 113, you wrote: "Because the high-fidelity radiation transport simulations described in the previous section are computationally very intense, we emulated the detector response as a function of the NPSM parameters using a machine learning trained vector-valued PCE surrogate model [43]." Please describe the advantages of using the surrogate model in terms of time and disc space saved considering the size of the problem. How many individual simulations would be required with the forward simulation model to solve that inversion problem? How much faster the surrogate model is compared to the forward simulation model?

Question 7

Starting on line 116, you wrote: "We performed the Bayesian inversion on the ⁶⁰Co (activity $A = 3.08(5) \times 10^5$ Bq) spectral dataset [30] leaving the remaining measurements for validation." ⁶⁰Co has two gamma-ray lines relatively close to each other. Would this feature affect the resulting Compton edge region, and consequently the accuracy of the NPSM? Is using ⁶⁰Co for calibration really a good choice? Please justify.

Question 8

Starting on line 129, you wrote: "We observe a shift of the Compton edge toward smaller spectral energies for an increase in $dE/ds|_{\text{Birks}}$ and η_e/h as well as a decrease in $dE/ds|_{\text{Trap}}$." And similarly, starting on line 194, you wrote: "In line with these observations, the non-proportionality is enhanced by a large e^-/h fraction, an increased Birks mechanism as well as a reduction in the e^-/h trapping rate [20, 24, 49]." In these two sentences, the statement about the two first parameters make sense, but the statement about the third one ($dE/ds|_{\text{Trap}}$) is counterintuitive. One could actually think the opposite, that is, that a reduction in the e^-/h trapping rate would make the system more "proportional" by leaving more e^-/h pairs available. Please explain why the non-proportionality is enhanced by a reduction in the e^-/h trapping rate.

Question 9

The section starting on line 143 and entitled "Intrinsic resolution" needs to be expanded further because, as it reads right now, its relevance for this manuscript doesn't manifest satisfactorily, and essential details ensuring sound understanding are omitted. Please consider answering the following questions while revising the content of this section.

What do you mean exactly by intrinsic resolution and empirical resolution? Resolution, especially intrinsic resolution, can be understood in different ways. Please clarify what is taken into account, and what is not, in each one.

How the intrinsic data are calculated? As they derive from new implementations, please provide more details about how the forward simulation model was used to create the dataset mentioned on line 147 (“[...] this dataset, we then trained [...]”). In particular, which data from the simulation results were extracted, and how those data were processed?

A brief summary about how the empirical data were collected and about how the empirical resolution model was derived would be great for completeness (I understand that those details are already published).

Please explain the importance of calculating the intrinsic resolution, and the importance of discussing it in this manuscript.

Please explain the importance of comparing the intrinsic resolution to the empirical resolution. What do we learn from this comparison? What are the impacts of this comparison? Why should we care about those results?

Question 10

Starting on line 164, you wrote: “We used the ^{88}Y ($A = 6.83(14) \times 10^5 \text{ Bq}$) and ^{137}Cs ($A = 2.266(34) \times 10^5 \text{ Bq}$) radiation measurements to validate our calibrated NPSM.” Moreover, the associated graphs in Fig. 4b and Fig. 4c show good improvements when using the NPSM to generate the simulated data. This validation suggests that the calibrated NPSM applies also for Compton edges found in energy windows ($\sim 375\text{-}625 \text{ keV}$, $\sim 550\text{-}850 \text{ keV}$, and $\sim 1450\text{-}1750 \text{ keV}$) outside the one used for its calibration ($\sim 800\text{-}1100 \text{ keV}$). Please explain why. Are there energy limits beyond which the calibrated NPSM would not apply anymore?

Question 11

In Fig. 4 a, the Compton edge predicted by the Compton scattering theory is reported by a vertical dashed line. To which ^{60}Co gamma-ray line this Compton edge value is for? Please precise.

Question 12

Starting on line 171, you wrote: “We quantify the Compton edge shift between the prediction ECE according to the Compton scattering theory and the measured detector responses to be $\approx 20 \text{ keV}$ for all measurements highlighted in Fig. 4.” Can you explain why that shift is constant (i.e., not energy dependent)?

Question 13

Starting on line 303, you wrote: “We scored the energy deposition events in the scintillation crystals individually on an event-by-event basis using the custom user routine `usreu` together with the detect card.”, and starting on line 309, you wrote: “To implement the NPSM described by Eq. 1, we developed an additional user routine `comscw`. Similar to [1, 64], we weight each individual energy deposition event in the scintillator, point-like or along the charged particle track, by the scintillation light yield given in Eq. 1. The resulting simulated response is then rescaled to match the energy calibration models derived in

[30].” These sentences don’t provide enough information to understand exactly how the NPSM was implemented into the forward simulation model. As this manuscript focuses explicitly on the NPSM, please provide more details about that specific implementation. Moreover, in equation (1), how dE/ds is calculated exactly? How is the light yield translated into energy in the spectrum? Can you provide more details about comscw, in the supplementary documentation perhaps?

Hope this helps!

Best regards

Reviewer #3 (Remarks to the Author):

Dear Authors,

I have found your paper very interesting, thank you the opportunity to comment on your fine work. I have a few thoughts that I like to share:

1. Mention the values of the nonproportionality parameters (Birks, e/h , trapping) from S. Payne, S. Hunter, L. Ahle, N. Cherepy, and E. Swanberg, “Nonproportionality of Scintillator Detectors. III. Temperature Dependence Studies”, IEEE Transactions in Nuclear Science 61, 2771-2777 (2014). Considering the very different approaches we’ve used, the agreement is not too bad!
2. I believe that this work was motivated the discrepancies in your first paper at the Compton edge (Breitenmoser, 2022). But then why is the photopeak modeled so well? Please explain.
3. The comment "Nonproportionality effects are known to increase with crystal size" is not correct. But of course, the resolution degrades. Have you explicitly modeled the light collection non-uniformity (?), which is large for the size crystals you have studied.
4. It would be great if Fig. 3f were plotted on a linear scale as the FWHM with the nonproportionality contribution and the "everything else" contribution plotted separately. And then you can plot the square root of the sum of the squares as the observed resolution.
5. Out of ignorance in Bayesian mathematics, I wonder why the three Sobol index peaks in Fig. 3e do not correspond to larger variance in Fig. 3d...

Again, very fine work.

1 Response to Referee #1

We would like to thank the referee for the thoughtful feedback as well as for the suggestions on how to improve the manuscript. Below we address all the concerns and questions raised by the referee. A full list of all minor changes made in the revised manuscript and the revised Supplementary Information File can be found in Section 4.

Comment #1:

Results are presented for sodium iodide – NaI(Tl), which is a widely used scintillator with significant nonproportionality. The inversion robustly estimated light output model parameters, producing results qualitatively like that reported by Payne et al. (2014) for NaI(Tl). Payne et al reported results for a wide temperature range. Nevertheless, it might be instructive to include for comparison temperature-interpolated values with the results of the MCMC inversion.

Reply:

We thank the referee for bringing up this point. We concur with his/her assessment and have thus incorporated the recommended comparison of temperature interpolated literature values and our results, alongside with a corresponding discussion. The two added paragraphs in the manuscript read as follows (lines 151–155 and 252–255):

Furthermore, our results significantly differ from best-estimate literature values, which we obtained using linear temperature interpolation on a dataset provided by Payne and his co-workers, i.e. $\eta_{e/h} = 4.53 \times 10^{-1}$, $dE/ds|_{\text{Trap}} = 1.2 \times 10^1 \text{ MeVcm}^{-1}$ and $dE/ds|_{\text{Birks}} = 1.853 \times 10^2 \text{ MeVcm}^{-1}$ for an ambient temperature of 18.8 °C [1].

However, we found statistically significant differences between our results and best-estimate literature values as well as between the individual scintillation crystals themselves. These results corroborate the experimental findings of Hull and his co-workers [2] and underscore the criticality of the NPSM calibration for every individual detector system.

Comment #2:

Limiting the current study to NaI(Tl) has the advantage of focusing on an exposition of the methodology. However, it would be helpful to see how well the methods work for other scintillators. The current generation of scintillators, including lanthanum bromide, have low non-proportionality. It is not clear how sensitive the Compton-edge probing method would be to light output parameters in this case.

Reply:

We agree with the referee that the application of the proposed methodology to other inorganic scintillators would be very interesting from a scientific point of view. However, as already indicated by the referee, such an extension would exceed the scope of the present study. That said, the referee's comment made us realize that we did not properly discuss potential limitations of our methodology, when it comes to scintillators, which are known to exhibit low non-proportionality effects. We therefore added the following paragraph in the discussion section (lines 307–313):

However, it is important to note that our methodology relies on the observation of Compton edge shifts with a sufficient signal-to-noise ratio (SNR). We have shown that these shifts are influenced by the strength of scintillation non-proportionality of a given scintillator. As a result, scintillator materials that exhibit only a mild non-proportional scintillation response, e.g. LaBr₃(Ce) or YAP(Ce), may present challenges for the calibration of a NPSM due to the reduced SNR in the Compton edge shift. Further investigations are required to assess the applicability of our methodology to such scintillators.

We would like to add that the scintillator material is without question an important factor when it comes to the strength of the scintillation non-proportionality. However, as discussed in the introduction of our manuscript, other factors may play an equally important role, such as activator type and concentration, ambient temperature or crystal defects, among others. Therefore, statements about the scintillation non-proportionality of a scintillator solely based on its crystal lattice might be difficult or even misleading. Furthermore, we would like to emphasize that our study not only discusses the NPSM calibration aspects but provides also the framework for a Monte Carlo based forward modeling for inorganic scintillators. Consequently, the proposed methodology may also be used to make predictions on the Compton edge shift for any inorganic scintillator given a known or predefined set of NPSM parameters and thereby support the additional investigations mentioned above.

Comment #3:

In addition, shifts in the gamma-ray photopeak positions also provide clues to nonproportionality. In focusing exclusively on the edges, the authors are not using all the information available to them to characterize light yield.

Reply:

The referee is right. Our method could be extended not only to include the mentioned full energy peak shifts but also the shift of annihilation escape peaks [3, 4], X-ray escape peaks [5–7] and sum peaks [8, 9] or any combination of those features. However, such an extension would exceed the scope of the present study. The main goal of our study is to demonstrate that Compton edge probing combined with Monte Carlo simulations and Bayesian inversion can successfully infer NPSMs for NaI(Tl) inorganic scintillators. That said, the referee's comment made us realize that we did not properly discuss further extensions of our methodology. We therefore added the following paragraph in the discussion section (lines 313-316):

That said, the presented methodology can be readily adapted using Bayes's theorem to address low SNR cases more effectively by combining multiple Compton edge domains or by probing additional spectral features distorted by the non-proportional scintillation response.

For a more in-depth discussion on the question why Compton edge shifts are particularly useful for NPSM calibration, we refer the reader to comment #1 for referee #2, comment #2 for referee #3 as well as the newly added Supplementary Method S1.4 in the revised Supplementary Information File.

2 Response to Referee #2

We would like to thank the referee for the very detailed feedback as well as for the suggestions on how to improve the manuscript. Below we address all the concerns and questions raised by the referee. A full list of all minor changes made in the revised manuscript and the revised Supplementary Information File can be found in Section 4.

Comment #1:

Starting on line 26, you wrote: “Second, various studies stated the conjecture that specific spectral features such as the Compton edges are shifted and distorted as a result of the non-proportional scintillation response [4, 8, 10–12].” Why is this affecting the Compton edges more than other parts of the spectrum? Please explain.

Reply:

This is a very interesting question. We must emphasize in advance that, as discussed in the introduction section of our manuscript, the origin of scintillation non-proportionality and any resulting spectral effects caused by these processes are not yet well understood [16, 19, 20]. This includes any spectral distortions such as shifts in the Compton edge [4, 8, 10–12], X-ray escape peaks [5–7] or sum peaks [8, 9]. That said, after dwelling on this point for quite some time, we decided to investigate this question systematically by deriving a semi-analytical model to improve our understanding about the underlying physics. We added a comprehensive description of this semi-analytical model, along with the key conclusions derived from it in the Supplementary Method S1.4 in the revised Supplementary Information File. For the reader’s convenience, we summarize the main results here:

First, it is important to note that the scintillation non-proportionality affects the entire pulse-height spectrum. However, the magnitude of the resulting spectral shifts depends on the number and the initial kinetic energy of the generated secondary electrons inside the scintillation crystal for a specific event. Due to the decreasing trend in the relative light yield function for higher energies (cf. Supplementary Fig. S27(b) in the revised Supplementary Information File), the magnitude of the spectral shift increases for an increase in the kinetic electron energy. Because for a Compton edge event, i.e. a single Compton scattering event with maximum deflection (full-backscattering), the generated secondary electron possesses the highest kinetic energy, we expect to see the most significant shifts exactly for these events. For more details on this topic, please have a look at the Supplementary Method S1.4 in the revised Supplementary Information File.

Comment #2:

Starting on line 48, you wrote: “In this study, we propose Compton edge probing together with Bayesian inversion to infer and calibrate NPSMs.” If the NPSM is calibrated using data taken from the Compton edge region, is it viable only in this region? In other words, in the forward simulation model, is the NPSM systematically applied to all energies in the spectrum, or limited to energies falling in Compton edge regions? Please clarify.

Reply:

Because the NPSM is a global model for all energies, a calibrated NPSM using data taken from the Compton edge region can indeed be applied to the entire spectral domain. We demonstrate the robustness of our approach in Section "Bayesian calibrated NPSM simulations" in the revised manuscript. In this section, we demonstrate the predictive power of the calibrated NPSM for the entire spectral domain.

To answer the second question: The NPSM is applied to all free electrons and positrons generated during the Monte Carlo simulation inside the scintillator volume, as described in detail in Section "Monte Carlo simulations" as well as in Section "Forward modeling" in the revised manuscript. In that sense, it is a fundamental global model and thereby also valid over the entire spectral range of our detector system. To underline this point, we added the following sentence on the lines 396–397 in the revised manuscript:

Using our methodology, we get simulated pulse-height spectra that incorporate non-proportional effects across the entire spectral range of our detector system.

In addition, we added a small note between the lines 224 and 227 in the revised manuscript to emphasize the global character of the NPSM as follows:

In addition to the insights into the Compton edge dynamics as well as the intrinsic resolution, the Bayesian inferred NPSM in combination with our forward model offers also the possibility to predict the full spectral detector response for new radiation sources accounting for non-proportional scintillation effects over the entire spectral range of our detector system.

Comment #3:

Starting on line 67, you wrote: "The adopted spectrometer consisted of four individual 10.2 cm × 10.2 cm × 40.6 cm prismatic NaI(Tl) scintillation crystals." From this sentence and from Fig. 1a, I understand that this detector contains actually four crystals. Was the NPSM calibrated for each crystal individually, or after combining the crystals together? If the latter, as each crystal can be different, even just slightly, how is it affecting the calibrated NPSM? Please clarify.

Reply:

We thank the referee for this interesting question. To account for the sensitivity of the NPSMs on the activator concentration and other scintillation crystal specific properties, we developed two separate inversion pipelines. In the first approach, Bayesian inversion was carried out separately for each of the four crystals using their individual pulse-height spectra. In the second pipeline, we considered all four scintillation crystals as one integrated scintillator and performed the Bayesian inversion on the combined pulse-height spectra (sum channel). Subsequently, we will refer to these two approaches as the *single* and *sum* mode inversion pipelines, respectively.

In the submitted manuscript, we did not include the results from the single inversion pipeline. In order to prevent any ambiguities, we have decided to include detailed descriptions and associated results of both pipelines in the manuscript and the Supplementary Information File. This resulted in major changes both in the manuscript as well as in the Supplementary Information File. For the reader's convenience, we list here the main changes and results. A full list of all

minor changes made in the revised manuscript and the revised Supplementary Information File can be found in Section 4.

1. In the revised manuscript between the lines 117–124, we added the following paragraph to explain the motivation and details of the two inversion pipelines:

To account for the sensitivity of the NPSMs on the activator concentration and other scintillation crystal specific properties, we developed two separate inversion pipelines. In the first approach, Bayesian inversion is carried out separately for each of the four crystals, using their individual pulse-height spectra. In the second pipeline, we consider all four scintillation crystals as one integrated scintillator and perform the Bayesian inversion on the combined pulse-height spectra (sum channel). Subsequently, we will refer to these two approaches as the *single* and *sum* mode inversion pipelines. For both pipelines, we performed the Bayesian inversion on the ^{60}Co (activity $A = 3.08(5) \times 10^5$ Bq) spectral dataset [12] leaving the remaining measurements for validation.

2. In addition to the already reported point estimates obtained by the sum mode inversion pipelines, we added also averaged estimates for the single mode inversion pipeline in the revised manuscript on the lines 140–147. The corresponding paragraphs read now:

Following the sum mode inversion pipeline, we present the solution to our inversion problem as a multivariate posterior distribution estimate in Fig. 2. We find a unimodal solution with a maximum a posteriori (MAP) probability estimate given by $\eta_{e/h} = 5.96^{+0.10}_{-0.17} \times 10^{-1}$, $dE/ds|_{\text{Trap}} = 1.46^{+0.03}_{-0.31} \times 10^1$ MeVcm $^{-1}$ and $dE/ds|_{\text{Birks}} = 3.22^{+0.46}_{-0.44} \times 10^2$ MeVcm $^{-1}$, where we used the central credible intervals with a probability mass of 95% to estimate the associated uncertainties. Combining the individual multivariate posterior distribution estimates from the single mode inversion pipeline, we obtain statistically consistent estimates, i.e. $\eta_{e/h} = 5.87^{+0.24}_{-0.20} \times 10^{-1}$, $dE/ds|_{\text{Trap}} = 1.41^{+0.17}_{-0.15} \times 10^1$ MeVcm $^{-1}$ and $dE/ds|_{\text{Birks}} = 3.17^{+1.11}_{-0.82} \times 10^2$ MeVcm $^{-1}$.

As reported, we did not find any significant difference between the averaged point estimates from the single mode inversion pipeline and the values from the sum mode inversion pipeline.

3. However, as expected, we found statistically significant differences between the point estimates for the individual scintillation crystals obtained by the single mode inversion pipeline. We include this finding in the revised manuscript on the lines 150–151 as follows:

However, we find statistically significant differences between the posterior point estimators for the individual scintillation crystals (Supplementary Table S2).

As written, we complemented the posterior statistics in the Supplementary Table S2 with the single inversion pipeline results in the revised Supplementary Information File. In addition, we added trace plots (Supplementary Figs. S2–S5), posterior point estimator convergence graphs (Supplementary Figs. S7–S10) as well as posterior distribution plots (Supplementary Figs. S11–S14) obtained by the single mode inversion pipeline.

4. We did not find any statistically significant difference between the two inversion pipelines for the Sobol sensitivity indices. We state this finding on the lines 169–170 in the revised manuscript as follows:

We get consistent results for a Hoeffding-Sobol sensitivity analysis of the individual scintillation crystals (Supplementary Fig. S17).

As written, we included total Sobol indices graphs obtained by the single mode inversion pipeline in the revised Supplementary Information File (Supplementary Fig. S17).

5. We did not find any statistically significant difference between the two pipelines for the posterior predictive distributions, i.e. the spectral Compton edge predictions. We state this finding on the lines 175–180 in the revised manuscript as follows:

We get consistent results using the single mode inversion pipeline (Supplementary Fig. S15). Furthermore, by comparing the posterior Compton edge predictions for the sum channel, we find no statistically significant difference between the two inversion pipelines (Supplementary Fig. S16). From a modeling perspective, it is interesting to add that we observe no significant difference for Compton edge predictions using the various point estimators discussed in the previous section for both inversion pipelines.

As written, we added two additional figures in the revised Supplementary Information File, i.e. Supplementary Fig. S15 with posterior Compton edge predictions for the individual crystals obtained by the single mode inversion pipeline as well as a spectral comparison of the sum and single mode inversion pipelines in Supplementary Fig. S16.

Comment #4:

The plots of $dE/ds|_{\text{Trap}}$ at the bottom of Fig. 2 and in Fig. S1 c, show long tails and wide variations, respectively, all ranging from 10 to 15 MeV cm⁻¹. Would that be due to too wide priors, to an omitted physical process in the NPSM, or to the trapping process not being fully represented in the Compton edge region, so the Bayesian inversion method can't quantify that parameter with the same level of precision as the two other parameters? What are your thoughts about this?

Reply:

The distribution mentioned in Fig. 2 shows samples from the multivariate posterior. In a Bayesian statistics framework, this quantity reflects the epistemic uncertainty about the statistical parameters given some set of data. We agree with the referee that the constraint on the $dE/ds|_{\text{Trap}}$ parameter seems indeed to be less tight than for the other parameters. However, as shown in the bottom-right diagonal panel in Fig. 2, the univariate marginal posterior distribution is still unimodal with a pronounced peak indicating a robust constraint also on this parameter. It is interesting to note that the constraints for $dE/ds|_{\text{Trap}}$ obtained by the single mode inversion pipeline appear to be smaller compared to the sum mode inversion pipeline (cf. Supplementary Table S2 and Supplementary Figs. S11–S14 in the revised Supplementary Information File). We conclude that the reason for the wider posterior distribution for $dE/ds|_{\text{Trap}}$ can be attributed to the model bias introduced by performing the inversion on the sum channel, in addition to the factors already noted by the referee. As discussed in the previous comment, this bias is statistically insignificant both for the posterior parameter estimates as well as for the posterior predictive distributions.

Coming to the second figure: The distribution in Fig. S1(c) reflects simply the sequential evolution of the samples drawn from the multivariate posterior probability density using MCMC. For this trace plot, we expect to see stationarity for the individual chains as well as good mixing between them after the burn-in phase. Both is fulfilled for $dE/ds|_{\text{Trap}}$.

Comment #5:

For example, starting on line 111, you wrote: “In addition, we fixed the Onsager related stopping power parameter $dE/ds|_{\text{Ons}}$ to 36.4 MeVcm^{-1} as suggested by previous investigators [1, 13].”, and you provided the value of the other parameters in Table S1. How would you explain that the parameters in equation (1) are all constant in energy (i.e., not energy dependent)?

Reply:

As discussed in Section “Forward modeling” in the manuscript, the sequence of the scintillation processes in inorganic scintillators can be qualitatively divided into five steps [16–18]:

1. Interaction of the ionizing radiation with the scintillator (radiation interaction)
2. Production of numerous secondary electrons, photons and plasmons (e^-e^- relaxation)
3. Electron–phonon relaxation producing excitation carriers (thermalization)
4. Migration of the generated excitation carriers to the luminescence centers and subsequently excitation of these centers (energy transfer)
5. Radiative relaxation of the excited luminescence centers (light emission)

The adopted mechanistic NPSM comprises now only the last three processes. The first two are simulated directly in our Monte Carlo simulations. As a result, all the parameters of the NPSM reflect physical processes after thermalization of the secondary particles, i.e. generation and transport of excitation carriers and not of the secondary particles. Consequently, these processes and thereby also the corresponding parameters can be regarded as statistically independent with respect to the energy of the secondary particles. To highlight these features of our NPSM, we added the following note on the lines 104 to 108 in the revised manuscript

As a result, all the parameters of the NPSM reflect physical processes after thermalization of the secondary particles, i.e. generation and transport of excitation carriers. Consequently, these processes and thereby also the corresponding parameters can be regarded as statistically independent with respect to the energy of the secondary particles.

For more details on this topic, the reader is referred to the books by Lecoq et al. [17] and Rodnyi [18].

Comment #6:

Starting on line 113, you wrote: “Because the high-fidelity radiation transport simulations described in the previous section are computationally very intense, we emulated the detector response as a function of the NPSM parameters using a machine learning trained vector-valued PCE surrogate model [14].” Please describe the advantages of using the surrogate model in terms of time and disc space saved considering the size of the problem. How many individual

simulations would be required with the forward simulation model to solve that inversion problem? How much faster the surrogate model is compared to the forward simulation model?

Reply:

We agree with the referee that we did not adequately motivate and discuss the advantages of applying the surrogate model. Therefore, we added a dedicated paragraph to discuss this topic in detail in section "Bayesian inversion" on the lines 133–139 in the revised manuscript. This paragraph reads as follows:

The surrogate model has excellent evaluation speed $\mathcal{O}(10^{-4}$ s) on a local workstation compared to $\mathcal{O}(10^3$ s) required for a single Monte Carlo simulation with sufficient precision on a computer cluster. Correspondingly, the surrogate model provides a significant acceleration of our Bayesian inversion computations, reducing their processing time by a factor of 10^7 . Considering the minimum number of forward model evaluations needed for a single Markov chain (Methods), the evaluation time can be reduced from $\mathcal{O}(10^2$ d) to $\mathcal{O}(1$ s).

In addition, we added more details on our computational resources in the methods section on the lines 383–385 in the revised manuscript:

All simulations were performed on a local computer cluster (7 nodes with a total number of 520 cores at a nominal clock speed of 2.6 GHz) at the Paul Scherrer Institute utilizing parallel computing.

The minimum number of forward model evaluations is $\mathcal{O}(10^4)$ for our problem (cf. trace plots in Supplementary Information File). It is worth mentioning that the evaluation speed for the Monte Carlo simulation might be significantly slower for more complex detector systems, e.g. satellite probes or particle physics experiments. Using proper scoring functions in combination with online NPSM evaluation, disc space on the other hand is of less importance for those kind of problems, i.e. both Monte Carlo simulation as well as the surrogate model data products require only minor disc space $\leq \mathcal{O}(10^1$ GB).

Comment #7:

Starting on line 116, you wrote: "We performed the Bayesian inversion on the ^{60}Co (activity $A = 3.08(5) \times 10^5$ Bq) spectral dataset [12] leaving the remaining measurements for validation." ^{60}Co has two gamma-ray lines relatively close to each other. Would this feature affect the resulting Compton edge region, and consequently the accuracy of the NPSM? Is using ^{60}Co for calibration really a good choice? Please justify.

Reply:

We thank the referee for this interesting question. There are two main reasons why we selected the ^{60}Co dataset for calibration. First, as shown in [12], the systematic uncertainties in the mass model are smallest for this dataset. Second, because ^{60}Co contains two Compton edges in close proximity, we can assess the robustness of our methodology for this somewhat more complex case compared to the other available datasets.

That said, the main focus during calibration remains with the Compton edge at 963.419(3) keV associated with the lower emission line at 1173.228(3) keV. The upper Compton edge at 1118.101(4) keV associated with the emission line at 1332.492(4) keV is heavily obscured by the lower full energy peak (Supplementary Fig. S15 in the revised Supplementary Information File). However, the small dip in the count rate at about 1080 keV for the single channel related to crystal 2 (Supplementary Fig. S15) and consequently also for the sum channel (Fig. 3 and 4) might actually be explained by the upper Compton edge. As discussed in the revised manuscript as well as in the Supplementary Method S1.4 in the revised Supplementary Information File, the upper Compton edge is also subjected to a scintillation non-proportionality induced spectral shift towards lower energies, which is why we are able to detect it for some of the detector channels.

Due to the fact that our methodology equally well predicts the pulse height spectra and in particular the Compton edge domain for both the detector channels with and without the discussed dip, we conclude that the inversion pipelines are able to successfully calibrate the NPSM in a robust and reliable way. In summary, we could not observe any significant sensitivity on the presence or absence of the second Compton edge in the ^{60}Co dataset. However, we agree with the referee that this point is not properly addressed in the submitted manuscript. Therefore, we added the following paragraph in the revised manuscript on the lines 124–128:

^{60}Co possesses two main photon emission lines at 1173.228(3) keV and 1332.492(4) keV with corresponding Compton edges according to the Compton scattering theory (Methods) at 963.419(3) keV and 1118.101(4) keV, respectively. However, in this study, we will focus on the lower Compton edge at 963.419(3) keV, because the upper edge is heavily obscured by the FEP at 1173.228(3) keV.

In addition, we added more information on the individual Compton edges in the caption for Fig. 4 in the revised manuscript:

The zoomed-in subfigures highlight the Compton edge region and include also the Compton edge E_{CE} predicted by the Compton scattering theory [21], i.e. [477.334(3), 699.133(3), 963.419(3), 1611.77(1)] keV associated with the photon emission lines at [661.657(3), 898.042(3), 1173.228(3), 1836.063(3)] keV, respectively.

Comment #8:

Starting on line 129, you wrote: “We observe a shift of the Compton edge toward smaller spectral energies for an increase in $dE/ds|_{\text{Birks}}$ and $\eta_{e/h}$ as well as a decrease in $dE/ds|_{\text{Trap}}$.” And similarly, starting on line 194, you wrote: “In line with these observations, the non-proportionality is enhanced by a large e^-/h fraction, an increased Birks mechanism as well as a reduction in the e^-/h trapping rate [15–17].” In these two sentences, the statement about the two first parameters make sense, but the statement about the third one ($dE/ds|_{\text{Trap}}$) is counterintuitive. One could actually think the opposite, that is, that a reduction in the e^-/h trapping rate would make the system more “proportional” by leaving more e^-/h pairs available. Please explain why the non-proportionality is enhanced by a reduction in the e^-/h trapping rate.

Reply:

That is a very interesting question. The important point to note here is the fact that the Onsager mechanism and the trapping mechanism are coupled in a nonlinear way, i.e. the trapping can be interpreted as a screening mechanism on the Onsager mechanism as discussed in detail by Vasil'ev and Gekhtin [16]. In other words, an increase in the trapping rate damps the Onsager term in Eq. 1 in the manuscript and thereby reduces overall the non-proportional characteristics of the scintillation light yield function. Having this in mind, we conclude that the observed trends in Fig. 3(c) are indeed in line with the scintillation physics theory as stated in the Discussion section in the manuscript. To clarify this point, we added the following paragraph in the revised manuscript on the lines 108–112:

From a physics perspective, it is important to note that the Onsager and trapping mechanisms are coupled in a nonlinear way, whereas the Birks mechanism can be regarded as independent of the other mechanisms. As discussed in detail by Vasil'ev and Gekhtin [16], we may therefore interpret the trapping of e^-/h pairs as a screening mechanism on the Onsager term in Eq. 1.

Comment #9:

The section starting on line 143 and entitled “Intrinsic resolution” needs to be expanded further because, as it reads right now, its relevance for this manuscript doesn't manifest satisfactorily, and essential details ensuring sound understanding are omitted. Please consider answering the following questions while revising the content of this section. What do you mean exactly by intrinsic resolution and empirical resolution? Resolution, especially intrinsic resolution, can be understood in different ways. Please clarify what is taken into account, and what is not, in each one. How the intrinsic data are calculated? As they derive from new implementations, please provide more details about how the forward simulation model was used to create the dataset mentioned on line 147 (“[...] this dataset, we then trained [...]”). In particular, which data from the simulation results were extracted, and how those data were processed? A brief summary about how the empirical data were collected and about how the empirical resolution model was derived would be great for completeness (I understand that those details are already published). Please explain the importance of calculating the intrinsic resolution, and the importance of discussing it in this manuscript. Please explain the importance of comparing the intrinsic resolution to the empirical resolution. What do we learn from this comparison? What are the impacts of this comparison? Why should we care about those results?

Reply:

We agree with the referee's assessment. Therefore, we made some major changes in the manuscript to provide a more extensive discussion about the intrinsic resolution topic. For the reader's convenience, we list here all the major changes. A full list of all minor changes made in the revised manuscript and the revised Supplementary Information File can be found in Section 4.

1. We added the following three paragraphs in the revised manuscript between the lines 182–205 to motivate and discuss the details about the intrinsic resolution study:

As already mentioned in the introduction, the scintillation non-proportionality not only distorts the spectral features in the pulse-height spectra but deteriorates also the spectral

resolution of a scintillator detector. This contribution to the overall resolution due to the scintillation non-proportionality will be referred to as intrinsic resolution σ_{intr} in accordance with previous studies [22–28]. The intrinsic resolution is of great importance for two key reasons.

First, it sets a fundamental lower limit on the achievable spectral resolution for a given scintillator material, making it a crucial factor in the development of new scintillators. As an example, in 1991, the scintillator Lu_2SiO_5 (LSO) was developed as an alternative to other available options at that time, such as $\text{Bi}_4\text{Ge}_3\text{O}_{12}$ (BGO). However, the performance of LSO led to considerable confusion within the research community as LSO exhibits a light yield more than four times greater than that of BGO, yet their energy resolutions are comparable [19]. Consequently, the energy resolution for LSO was not dominated by counting statistics but some other factor. Thanks to the development of the Compton coincidence measurement technique in 1994 [29], subsequent experimental studies have conclusively shown that the pronounced scintillation non-proportionality of the LSO scintillator was indeed the underlying factor responsible for the observed resolution degradation [27, 30]. This example showcases the need for a better understanding and prediction of the intrinsic resolution in the development of new scintillators [31].

Second, from a more technical perspective, the intrinsic resolution is a key component in the postprocessing pipeline for Monte Carlo simulations including NPSMs (Methods). In the forward model discussed before, the transport of scintillation photons, signal amplification by the photomultiplier tube and subsequent signal postprocessing in the multichannel analyzer are not included. As a result, to account for the additional resolution degradation by these processes, we need to perform a spectral broadening operation using a dedicated energy resolution model based on the measured total energy resolution as well as the intrinsic contribution [8].

2. To highlight the importance and significance of the intrinsic resolution results as well as to discuss the main conclusions from the comparison of intrinsic and total resolution, we modified the corresponding paragraph in the Discussion section between the lines 274 and 284 in the revised manuscript as follows:

Using the Bayesian calibrated NPSM, we are also able to numerically characterize the contribution of the scintillation non-proportionality to the overall energy resolution. This intrinsic resolution sets a fundamental lower limit on the achievable spectral resolution for a given scintillator material, making it a key factor in the development of new scintillators. At higher photon energies ($E_\gamma \gtrsim 400$ keV), we observed a significant contribution to the total spectral resolution ($\geq 35\%$) with a maximum of $\approx 42\%$ around 440 keV. At lower energies (10 keV $\lesssim E_\gamma \lesssim 400$ keV), the intrinsic contribution is reduced and shows substantial distortions around the K-absorption edge for iodine at about 33 keV. We conclude that the non-proportional scintillation is a significant contributor to the total energy resolution of NaI(Tl). These observations are in good agreement with previous results [27, 28, 32–35] and thereby substantiate not only the predictive power of our numerical model but showcase also its potential as a novel tool in the development of new scintillators.

3. As proposed by the referee, we added a short summary about the total resolution σ_{tot} between the lines 466 and 484 in the Methods section:

In this last section, we will discuss the derivation of the energy resolution models adopted in this study. We start with the model to characterize the overall or total energy resolution σ_{tot} for our detector system and describe in a second step the derivation of the intrinsic resolution model σ_{intr} . It is important to note that in contrast to σ_{intr} , we provide here only a short summary of the key aspects involved in σ_{tot} . The entire postprocessing pipeline to derive σ_{tot} was already thoroughly discussed in a previous study [12]. For more details, we kindly refer the reader to the dedicated study.

For each scintillation crystal, we quantified σ_{tot} by characterizing the spectral dispersion of measured FEPs associated with known photon emission lines from specific radionuclides. The corresponding pipeline can be divided into three steps. In a first step, we extracted specific spectral domains containing a singlet or multiplet of FEPs from a set of measured count rate spectra covering a spectral range between 122 and 1836 keV [36]. In a second step, we fitted a spectral peak model based on a sum of independent Gaussian peaks together with a numerical baseline [37] to the selected singlets or multiplets using weighted non-linear least-squares (WNLLS) regression combined with the interior-reflective Newton method [38]. In the third step, we extracted the Gaussian standard deviation parameters from the fitted FEPs as a characteristic measure for the spectral resolution. By combining these empirical resolution values with the known emission line energies, we derived an exponential model to describe σ_{tot} as a function of the photon energy E_γ adopting again WNLLS. The resulting relative generalization error, characterized by leave-one-out cross-validation, is $< 0.2\%$ for all scintillation crystals.

Moreover, we extended the paragraphs explaining the technical details about the intrinsic resolution analysis in the same Methods section between the lines 485 and 511 as follows:

To derive a model for σ_{intr} , we performed in a first step additional Monte Carlo simulations for an isotropic and uniform monoenergetic photon flux of energy $10 \text{ keV} \leq E_\gamma \leq 3200 \text{ keV}$. To account for the different spectral scales, we applied a non-uniform experimental design for the photon energy E_γ with a 2 keV spacing below 110 keV and 100 keV spacing above. Moreover, to account for the non-proportional scintillation physics, we ran all simulations with the Bayesian calibrated NPSM, i.e. the derived MAP point estimators. The mass model for those simulations features a $10.2 \text{ cm} \times 10.2 \text{ cm} \times 40.6 \text{ cm}$ prismatic NaI(Tl) scintillation crystal embedded in a vacuum environment. In a second step, we extracted the mean light yield values from the simulated FEPs (Supplementary Algorithm S1). Similar to the measured spectra, we can then derive a simple polynomial energy calibration model using WNLLS to convert the simulated light yield to energy [12]. In a third step, we adopted the extracted σ_{intr} from the individual energy calibrated FEPs to train a Gaussian Process (GP) regression model with [39]: [...] It is worth noting that, due to the known asymmetry in the FEPs [8, 23, 27], we adopted numerical estimates both for the mean and standard deviation parameters associated with the individual FEPs. [...] It is important to add that in case of the experimental design \mathcal{X} adopted to train the surrogate model, we ran the pipeline for σ_{intr} with the corresponding set of NPSM parameters defined by \mathcal{X} .

Last but not least, we incorporated an additional paragraph discussing the methodology to perform the broadening operation mentioned in the Results section between the lines 512 and 521:

As discussed already in the Results section, the intrinsic resolution is also a key component in the postprocessing pipeline for Monte Carlo simulations including NPSMs. Because the Monte Carlo simulations performed for the forward model only inherently include the intrinsic resolution, we need to perform a spectral broadening operation to account for the additional energy resolution degradation due to the transport of scintillation photons, signal amplification by the photomultiplier tube and subsequent signal postprocessing in the multichannel analyzer. Similar to [8], we assume statistical independence between the resolution degradation due to the scintillation non-proportionality and the aforementioned neglected processes in the Monte Carlo simulations. We can then perform the broadening operation as described in [12] with an adapted dispersion $\sqrt{\sigma_{\text{tot}}^2 - \sigma_{\text{intr}}^2}$. For completeness, we included this adapted dispersion model in Fig. 3f.

Note that we changed the subtitle for this entire subsection from "Intrinsic resolution modelling" to "Resolution modeling" to account for the broader information content discussed in this section. In addition, we changed the photon energy symbol from E to E_γ for all resolution related discussions to avoid any ambiguity with the spectral, deposited or kinetic electron energies. Moreover, as discussed in the comment #4 for referee #3, we changed Fig. 3(f) slightly (linear scaling and additional line to highlight adapted dispersion model).

Comment #10:

Starting on line 164, you wrote: "We used the ^{88}Y ($A = 6.83(14) \times 10^5$ Bq) and ^{137}Cs ($A = 2.266(34) \times 10^5$ Bq) radiation measurements to validate our calibrated NPSM." Moreover, the associated graphs in Fig. 4b and Fig. 4c show good improvements when using the NPSM to generate the simulated data. This validation suggests that the calibrated NPSM applies also for Compton edges found in energy windows (375-625 keV, 550-850 keV, and 1450-1750 keV) outside the one used for its calibration (800-1100 keV). Please explain why. Are there energy limits beyond which the calibrated NPSM would not apply anymore?

Reply:

The referee's conjecture is correct. As already discussed in the comments #2, #7 and #8, the NPSM is a global model based on the differential stopping power of the generated secondary electrons within the scintillator. We expect therefore a statistically significant improvement in the Compton edge region not only for the Bayesian calibrated spectrum, i.e. the selected ^{60}Co Compton edge domain, but also for other domains not seen before, such as the ones for ^{88}Y and ^{137}Cs . As correctly observed by the referee and discussed in detail in section "Bayesian calibrated NPSM simulations", we find indeed a significant improvement in the Compton edge prediction substantiating the predictive power of our numerical model. To underline this point, we added the following sentence on the lines 396–397 in the revised manuscript (as already discussed in comment #2):

Using our methodology, we get simulated pulse-height spectra that incorporate non-proportional effects across the entire spectral range of our detector system.

In addition, we added a small note between the lines 224 and 227 in the revised manuscript to emphasize the global character of the NPSM as follows:

In addition to the insights into the Compton edge dynamics as well as the intrinsic resolution, the Bayesian inferred NPSM in combination with our forward model offers also the possibility to predict the full spectral detector response for new radiation sources accounting for non-proportional scintillation effects over the entire spectral range of our detector system.

Regarding the question about the energy limits: From a computational point of view, the adopted Monte Carlo code FLUKA has thoroughly benchmarked transport models for a variety of particles with very broad energy ranges, e.g. electrons can be transported for an energy range of [1 keV, 1 PeV] [40]. The same is true for photons and charged hadrons with energy ranges of [100 eV, 10 PeV] and [1 keV, 20 TeV], respectively. However, the NPSM implemented in this study might be more restrictive in terms of the energy range, because it is based on a mechanistic approach. Validation measurements were performed up to electron kinetic energies ≈ 500 keV using the Compton coincidence measurement technique [13, 41]. On the other hand, because the NPSM characterizes only processes after thermalization as discussed already in comment #5, no additional systematic errors are expected for higher primary particle energies. This is also supported by the observed trends in the Supplementary Figs. S28 and S29 in the revised Supplementary Information File, i.e. we find decreasing effects of the NPSM on the mean Compton edge shifts and the mean number of Compton scatter events before absorption for higher photon energies (cf. also comment #1). So in short, we believe that the proposed methodology can be applied also for more exotic particle energies. To prove this, further investigations are required. This is however beyond the scope of the current study.

Comment #11:

In Fig. 4 a, the Compton edge predicted by the Compton scattering theory is reported as a vertical dashed line. To which ^{60}Co gamma-ray line this Compton edge value is for? Please precise.

Reply:

As already mentioned in comment #7, we added the suggested information in the caption for Fig. 4 in the revised manuscript:

The zoomed-in subfigures highlight the Compton edge region and include also the Compton edge E_{CE} predicted by the Compton scattering theory [21], i.e. [477.334(3), 699.133(3), 963.419(3), 1611.77(1)] keV associated with the photon emission lines at [661.657(3), 898.042(3), 1173.228(3), 1836.063(3)] keV, respectively.

For additional changes regarding the Compton edge energies in particular for ^{60}Co , the reader is kindly referred to comment #7.

Comment #12:

Starting on line 171, you wrote: “We quantify the Compton edge shift between the prediction E_{CE} according to the Compton scattering theory and the measured detector responses to be ≈ 20 keV for all measurements highlighted in Fig. 4.” Can you explain why that shift is constant (i.e., not energy dependent)?

Reply:

In accordance with our new semi-analytical model discussed in comment #1, we had to revise our statement about the Compton edge trend with energy on the lines 235–236 in the revised manuscript as follows:

In line with the Compton scatter theory (Supplementary Methods S1.4), we find a moderate increase in the Compton edge shift for higher photon energies.

A thorough description of the revised analysis and quantification of the Compton edge shift as well as the already discussed semi-analytical model is added in the Supplementary Information File (Supplementary Method S1.4).

Comment #13:

Starting on line 303, you wrote: “We scored the energy deposition events in the scintillation crystals individually on an event-by-event basis using the custom user routine *usreou* together with the *detect* card.”, and starting on line 309, you wrote: “To implement the NPSM described by Eq. 1, we developed an additional user routine *comscw*. Similar to [8, 9], we weight each individual energy deposition event in the scintillator, point-like or along the charged particle track, by the scintillation light yield given in Eq. 1. The resulting simulated response is then rescaled to match the energy calibration models derived in [12].” These sentences don’t provide enough information to understand exactly how the NPSM was implemented into the forward simulation model. As this manuscript focuses explicitly on the NPSM, please provide more details about that specific implementation. Moreover, in equation (1), how dE/ds is calculated exactly? How is the light yield translated into energy in the spectrum? Can you provide more details about *comscw*, in the supplementary documentation perhaps?

Reply:

We thank the referee for this suggestion. First, we would like to emphasize that all user routines including *comscw* are accessible via the ETH research data repository <https://doi.org/10.3929/ethz-b-000595727> [42]. We have to add that the access to these routines needs to be requested by e-mail, in order to comply with the FLUKA license. This in mind, we agree with the referee that additional information in the Supplementary Information File might be beneficial for some readers. Therefore, we decided to include a pseudo-code in the Supplementary Information File (Supplementary Algorithm S1) to highlight the algorithm implemented in *comscw* in more details. In addition, we added cross-references to the supplementary algorithm S1 in the revised manuscript between the lines 394 and 395 as well as on line 493.

Regarding the question of the light yield to energy conversion: We added more information about this conversion in the revised methods section between the lines 485 and 511 as already discussed in the comment #9 (point 3). For the reader’s convenience, we repeat here some of the statements. For more information, please refer to comment #9.

To derive a model for σ_{intr} , we performed in a first step additional Monte Carlo simulations for an isotropic and uniform monoenergetic photon flux of energy $10 \text{ keV} \leq E_\gamma \leq 3200 \text{ keV}$. To account for the different spectral scales, we applied a non-uniform experimental design for the photon energy E_γ with a 2 keV spacing below 110 keV and 100 keV spacing above. Moreover, to

account for the non-proportional scintillation physics, we ran all simulations with the Bayesian calibrated NPSM, i.e. the derived MAP point estimators. The mass model for those simulations features a $10.2 \text{ cm} \times 10.2 \text{ cm} \times 40.6 \text{ cm}$ prismatic NaI(Tl) scintillation crystal embedded in a vacuum environment. In a second step, we extracted the mean light yield values from the simulated FEPs (Supplementary Algorithm S1). Similar to the measured spectra, we can then derive a simple polynomial energy calibration model using WNLLS to convert the simulated light yield to energy [12]. In a third step, we adopted the extracted σ_{intr} from the individual energy calibrated FEPs to train a Gaussian Process (GP) regression model with [39]: [...] It is worth noting that, due to the known asymmetry in the FEPs [8, 23, 27], we adopted numerical estimates both for the mean and standard deviation parameters associated with the individual FEPs.

3 Response to Referee #3

We would like to thank the referee for the thoughtful feedback as well as for the suggestions on how to improve the manuscript. Below we address all the concerns and questions raised by the referee. A full list of all minor changes made in the revised manuscript and the revised Supplementary Information File can be found in Section 4.

Comment #1:

Mention the values of the nonproportionality parameters (Birks, e/h , trapping) from S. Payne, S. Hunter, L. Ahle, N. Cherepy, and E. Swanberg, "Nonproportionality of Scintillator Detectors. III. Temperature Dependence Studies", IEEE Transactions in Nuclear Science 61, 2771-2777 (2014). Considering the very different approaches we've used, the agreement is not too bad!

Reply:

We agree with the referee. However, as discussed in comment #1 for referee #1, the differences between Payne's and our results are still statistically significant. This is expected as discussed in the Introduction and Discussion sections, i.e. we suspect that these differences can be mainly attributed to the known sensitivity on the activator concentration as well as crystal defects. These results corroborate the experimental findings of Hull and his co-workers [2] and underscore the criticality of the NPSM calibration for every individual detector system.

Comment #2:

I believe that this work was motivated the discrepancies in your first paper at the Compton edge (Breitenmoser, 2022). But then why is the photopeak modeled so well? Please explain.

Reply:

This is a very interesting question. There are mainly three reasons. First, we expect the largest spectral distortions for Compton edges as discussed in comment #1 for referee #2. For the reader's convenience, we repeat here again this discussion:

We must emphasize in advance that, as discussed in the introduction section of our manuscript, the origin of scintillation non-proportionality and any resulting spectral effects caused by these processes are not yet well understood [16, 19, 20]. This includes any spectral distortions such as shifts in the Compton edge [4, 8, 10–12], X-ray escape peaks [5–7] or sum peaks [8, 9]. That said, after dwelling on this point for quite some time, we decided to investigate this question systematically by deriving a semi-analytical model to improve our understanding about the underlying physics. We added a comprehensive description of this semi-analytical model, along with the key conclusions derived from it in the Supplementary Method S1.4 in the revised Supplementary Information File. For the reader's convenience, we summarize the main results here:

First, it is important to note that the scintillation non-proportionality affects the entire pulse-height spectrum. However, the magnitude of the resulting spectral shifts depends on the number and the initial kinetic energy of the generated secondary electrons inside the scintillation crystal for a specific event. Due to the decreasing trend in the relative light yield function for higher

energies (cf. Supplementary Fig. 27(b) in the revised Supplementary Information File), the magnitude of the spectral shift increases for an increase in the kinetic electron energy. Because for a Compton edge event, i.e. a single Compton scattering event with maximum deflection (full-backscattering), the generated secondary electron possesses the highest kinetic energy, we expect to see the most significant shifts exactly for these events. For more details on this topic, please have a look at the Supplementary Method S1.4 in the revised Supplementary Information File.

Secondly, the significant resolution degradation of NaI(Tl) due to various processes such as the transport of scintillation photons, signal amplification by the photomultiplier tube and subsequent signal postprocessing in the multichannel analyzer obscures the non-proportional scintillation effects on the photopeaks, e.g. asymmetric peak shapes. This can be seen in Fig. 3(f) by the ratio $\sigma_{\text{intr}}^2/\sigma_{\text{tot}}^2$, i.e. we have always $\sigma_{\text{intr}}^2/\sigma_{\text{tot}}^2 < 0.5$. In other words, the resolution degradation due to processes other than scintillation non-proportionality is still significant.

Last but not least, as discussed in the Supplementary Method S1.4, we performed the energy calibration based on the photopeaks. Therefore, energy shifts are only observable with respect to other spectral features such as Compton edges or sum peaks. In other words, spectral shifts cannot be analyzed in isolation but need to be related to the photopeaks. For more information, we kindly refer to the dedicated discussion in the Supplementary Method S1.4.

Comment #3:

The comment "Nonproportionality effects are known to increase with crystal size" is not correct. But of course, the resolution degrades. Have you explicitly modeled the light collection non-uniformity (?), which is large for the size crystals you have studied.

Reply:

We respectfully disagree with the referee that the statement "Nonproportionality effects are known to increase with crystal size" is incorrect. As correctly stated by the referee, the resolution or more specifically the intrinsic resolution, which is a direct consequence of the scintillation nonproportionality (cf. comment #9 for referee #2), increases with crystal size. Hence, we may state that the "nonproportionality effect" in form of the intrinsic resolution increases with increasing crystal size. This finding was verified independently by multiple experimental studies [8, 23, 27, 43].

Furthermore, we investigate the magnitude of the Compton edge shift as a function of the scintillation crystal size in the Supplementary Method S1.4 and in particular Fig. S29(b). We find a pronounced and consistent increase in the Compton edge shift for an increase in crystal size over the entire spectral domain 500 keV to 2000 keV. Consequently, we conclude that the spectral distortions due to scintillation non-proportionality, in particular Compton edge shifts, increase with crystal size. From our semi-analytical model and the results in Fig. S29(a), it is evident that this trend for Compton edge shifts can be explained by the increase in the number of Compton scatter events before absorption for bigger scintillation crystals. Of course, these results need to be validated with accurate Compton edge shift measurements for different scintillator sizes using proper coincidence measurement techniques as discussed in the Supplementary Method S1.4. This exceeds the scope of the present work, but will be investigated further in future studies.

Regarding the light collection non-uniformity: We account for this non-uniformity indirectly by performing a broadening operation using a dedicated resolution model. To discuss this point more thoroughly, we performed a major revision of the corresponding Methods section between the lines 466 and 521. We kindly refer to our related reply in comment #9 for referee #2 for more information on this topic.

Comment #4:

It would be great if Fig. 3f were plotted on a linear scale as the FWHM with the nonproportionality contribution and the "everything else" contribution plotted separately. And then you can plot the square root of the sum of the squares as the observed resolution.

Reply:

As suggested, we revised Fig. 3(f). It includes now a linear scale for both the relative resolution as well as the resolution ratio. In addition, we added the "everything else" contribution in the plot, i.e. $\sqrt{\sigma_{\text{tot}}^2 - \sigma_{\text{intr}}^2}$ in combination with a corresponding cross-reference in the method section on the lines 520 to 521. Note that we keep σ_{tot} because we want to avoid any ambiguity, which variables were actually measured and which were calculated. In this study, we determined σ_{tot} as well as σ_{intr} and based on these values computed the "everything else" contribution. Furthermore, due to the asymmetry in the intrinsic FEPs, we do not replace σ by the FWHM [27]. Last but not least, we have to add that we revised also the resolution ratio slightly, i.e. we changed it from $\sigma_{\text{intr}}/\sigma_{\text{tot}}$ to $\sigma_{\text{intr}}^2/\sigma_{\text{tot}}^2$ in accordance with our revised discussion in the Methods section between the lines 466 and 521.

Comment #5:

Out of ignorance in Bayesian mathematics, I wonder why the three Sobol index peaks in Fig. 3e do not correspond to larger variance in Fig. 3d...

Reply:

This is another interesting question. The reason for the three peaks in Fig. 3(e) can be attributed to the specific definition of the total Sobol indices related to the global variance decomposition theory derived by Sobol [44]. As discussed in the Supplementary Method S1.2, the total Sobol index S_i^T quantifies the total effect of an input parameter X_i on a model $\mathcal{M}(\mathbf{X})$. In particular, it includes not only the effect of X_i alone but in addition the effect induced by all interactions between X_i and the other variables. This is also the reason, why the sum of the total Sobol indices exceeds 1 for some parts. As an example, if we have an interaction between the variables X_1 and X_2 , their interaction effect on \mathcal{M} is counted twice, once in S_1^T and another time in S_2^T . This example in mind, it is easy to see that the peaks highlight regions, where the interaction terms between the individual variables significantly contribute to the total effect on \mathcal{M} . We can also see that these points coincide with regions, where the prior predictive distribution, i.e. the probability density for the detector response for uniformly distributed input variables, has minimum variability. We have to add that for our conclusions, the absolute values of S_i^T are of less importance. In our study, we are more interested in the relative size of S_i^T , because the comparison of these values allows us to draw conclusions about the relative importance of the

corresponding variables for the model response. For the reader's convenience, we decided to include an additional paragraph in the supplementary section S1.2 in the revised Supplementary Information File to discuss this topic in more detail:

As a result, S_i^T includes not only the effect of X_i alone but in addition the effect induced by all interactions between X_i and the other variables. This is also the reason, why the sum of the total Sobol indices $\sum_i S_i^T$ can in fact exceed 1. As an example, if we have an interaction between the variables X_1 and X_2 , their interaction effect on \mathcal{M} is counted twice, once in S_1^T and another time in S_2^T . This example in mind, it is easy to see that the peaks in Fig. 3(e) in the main study highlight regions, where the interaction terms between the individual variables significantly contribute to the total effect on \mathcal{M} . We have to add that for our study, the absolute values of S_i^T are of less importance. We are more interested in the relative size of S_i^T , because the comparison of these values allows us to draw conclusions about the relative importance of the corresponding variables X_i for the model response $\mathcal{M}(\mathbf{X})$.

4 Summary of Changes

Here, we list all minor changes not discussed in the referee reports. We start with the changes in the revised manuscript:

1. For the reader's convenience, we deleted the symbols L and E on the line 12 in the Introduction section.
2. In line with the guide for authors, we changed the tense for the last paragraph in the Introduction from past to present.
3. In the section "Compton edge probing", we added a small note highlighting the individual read-out for the four scintillation crystals.
4. For the reader's convenience, we changed the font style for all code and algorithm names to typewriter style.
5. In line with the guide for authors, we revised the abbreviations in the entire manuscript and ensured that they are stated at the first time they appear as well as in each figure caption.
6. To highlight the specific inversion pipeline used to create the individual figures, we added corresponding notes in the captions for the Figs. 2–4 (cf. comment #3 for referee #2).
7. We found a small bug in one of our postprocessing pipeline, which led to a slight underestimation of the credible interval ranges. We corrected this bug and had to change therefore the following items in the manuscript:
 - the uncertainty values for the MAP point estimates on the lines 142 and 143
 - the prior and posterior distributions in Fig. 3(d)

It is important to add that none of the conclusions drawn from these results are affected by these changes.

8. We found a small typo in the Discussion section, i.e. we replaced the term e^-/h by $\eta_{e/h}$ on the line 266 in the revised manuscript.
9. For the reader's convenience, we deleted the symbols L and $-dE/ds$ on the line 301 in the Discussion section.
10. We added more details about the adopted detector system in the Methods section "Gamma-ray spectrometry" on the lines 337–339.
11. In line with our discussion in section S1.4 in the revised manuscript, we added a small note on the assumptions for Eq. 2 on line 360 in the revised manuscript.
12. Because of the increased size and supplementary character of the revised table containing information about the marginal priors, we decided to shift this table to the Supplementary Information File (Supplementary Table S1). A corresponding cross-reference is added in the revised manuscript on the line 452.
13. Due to the increased size of the Supplementary Information File, we added specific cross-references pointing to the individual supplementary items (lines 149–151, 170, 176, 178, 235–236, 245, 367, 371, 394–395, 435, 452, 461–463, 493) in the revised manuscript.
14. For the reader's convenience, we removed the grid lines in Fig. 4.

We continue with the changes related to the revision of the Supplementary Information File:

1. To account for the increasing size after revision, we added a Table of Contents, a List of Figures as well as a List of Tables in the beginning of the document.
2. As discussed in several comments, we added a new Supplementary Method Section S1.4 to discuss the Compton edge shift in more detail adopting a semi-analytical model.
3. As discussed in several comments, we added multiple new supplementary figures in the revised Supplementary Information File (Figs. S2–S5, S7–S17, S26–S29).
4. Similar to the revised manuscript, we added additional notes in each relevant caption with which inversion pipeline the figure or table was created.
5. For the reader's convenience, we removed the grid lines in Fig. S18.
6. Due to the found bug mentioned before, we corrected the credible intervals for the sum mode inversion pipeline in Table S2.
7. As already mentioned above, we extended the table containing the marginal prior information and shifted this table from the main manuscript to the revised Supplementary Information File (Table S1).
8. We adapted the supplementary tables S2 and S4 to include also the results from the single mode inversion pipeline and the individual crystals, respectively.
9. Together with the newly added Supplementary Method Section S1.4, we added also a new table S5 to list all the material properties adopted for the calculations in Section S1.4.
10. As discussed in the comment #13 for referee #2, we included a description of the `comscw` user routine by adding the supplementary algorithm S1 to the revised Supplementary Information File.

We conclude with a list of minor changes applied to both revised files.

1. For both revised files, we adapted the reference style in the Bibliography section to be in line with the journal's reference style:
 - We changed the font style for the term "et al." from italic to normal.
 - We shifted the year to the end of each reference entry.
 - We removed the journal issue/number as well as the doi entry for all articles.
 - We corrected some typos in the author names, e.g. we replaced S by Ś in the name "Świderski".
 - We exchanged the full journal names with their ISO4 abbreviations.
 - We ensured that the comma after the volume entry for articles appears in bold.
2. Furthermore, for consistency reasons, we changed the spelling of the following words in both revised files :
 - "modelling" to "modeling"
 - "can not" to "cannot"
 - "scintillator crystal" to "scintillation crystal"
 - "divided in" to "divided into"
 - "data set" to "dataset"

References

- [1] Payne, S. A., Hunter, S., Ahle, L., Cherepy, N. J. & Swanberg, E. Nonproportionality of scintillator detectors. III. Temperature dependence studies. *IEEE Trans. Nucl. Sci.* **61**, 2771–2777 (2014).
- [2] Hull, G. et al. Measurements of NaI(Tl) electron response: Comparison of different samples. *IEEE Trans. Nucl. Sci.* **56**, 331–336 (2009).
- [3] Zerby, C. D. & Moran, H. S. Calculation of the pulse-height response of NaI(Tl) scintillation counters. *Nucl. Instrum. Methods* **14**, 115–124 (1961).
- [4] Saito, K. & Moriuchi, S. Monte Carlo calculation of accurate response functions for a NaI(Tl) detector for gamma rays. *Nucl. Instrum. Methods* **185**, 299–308 (1981).
- [5] Collinson, A. J. & Hill, R. The Fluorescent Response of NaI(Tl) and CsI(Tl) to X Rays and γ Rays. *Proc. Phys. Soc.* **81**, 883–892 (1963).
- [6] Wayne, L. R., Heindl, W. A., Hink, P. L. & Rothschild, R. E. Response of NaI(Tl) to X-rays and electrons. *Nucl. Instrum. Methods Phys. Res. A* **411**, 351–364 (1998).
- [7] Khodyuk, I. V., Rodnyi, P. A. & Dorenbos, P. Nonproportional scintillation response of NaI:Tl to low energy x-ray photons and electrons. *J. Appl. Phys.* **107**, 113513 (2010).
- [8] Cano-Ott, D. et al. Monte Carlo simulation of the response of a large NaI(Tl)total absorption spectrometer for β -decay studies. *Nucl. Instrum. Methods Phys. Res. A* **430**, 333–347 (1999).
- [9] Rasco, B. C. et al. The nonlinear light output of NaI(Tl) detectors in the Modular Total Absorption Spectrometer. *Nucl. Instrum. Methods Phys. Res. A* **788**, 137–145 (2015).
- [10] Shi, H. X., Chen, B. X., Li, T. Z. & Yun, D. Precise Monte Carlo simulation of gamma-ray response functions for an NaI(Tl) detector. *Appl. Radiat. Isot.* **57**, 517–524 (2002).
- [11] Gardner, R. P. & Sood, A. A Monte Carlo simulation approach for generating NaI detector response functions (DRFs) that accounts for non-linearity and variable flat continua. *Nucl. Instrum. Methods Phys. Res. B* **213**, 87–99 (2004).
- [12] Breitenmoser, D., Butterweck, G., Kasprzak, M. M., Yukihiro, E. G. & Mayer, S. Experimental and Simulated Spectral Gamma-Ray Response of a NaI(Tl) Scintillation Detector used in Airborne Gamma-Ray Spectrometry. *Adv. Geosci.* **57**, 89–107 (2022).
- [13] Payne, S. A. et al. Nonproportionality of scintillator detectors: Theory and experiment. II. *IEEE Trans. Nucl. Sci.* **58**, 3392–3402 (2011).

- [14] Blatman, G. & Sudret, B. Sparse polynomial chaos expansions of vector-valued response quantities. *Proc. Int. Conf. ICOSSAR* (2013).
- [15] Beck, P. R. et al. Nonproportionality of Scintillator Detectors. V. Comparing the Gamma and Electron Response. *IEEE Trans. Nucl. Sci.* **62**, 1429–1436 (2015).
- [16] Vasil'Ev, A. N. & Gektin, A. V. Multiscale approach to estimation of scintillation characteristics. *IEEE Trans. Nucl. Sci.* **61**, 235–245 (2014).
- [17] Lecoq, P., Gektin, A. & Korzhik, M. *Inorganic Scintillators for Detector Systems* (Springer, Berlin, 2017).
- [18] Rodnyi, P. A. *Physical Processes in Inorganic Scintillators* (CRC Press, New York, 1997).
- [19] Moses, W. W., Payne, S. A., Choong, W. S., Hull, G. & Reutter, B. W. Scintillator non-proportionality: Present understanding and future challenges. *IEEE Trans. Nucl. Sci.* **55**, 1049–1053 (2008).
- [20] Moses, W. W. et al. The origins of scintillator non-proportionality. *IEEE Trans. Nucl. Sci.* **59**, 2038–2044 (2012).
- [21] Knoll, G. F. *Radiation Detection and Measurement* (John Wiley & Sons, New York, 2010).
- [22] Iredale, P. The effect of the non-proportional response of NaI(Tl) crystals to electrons upon the resolution for γ -rays. *Nucl. Instrum. Methods* **11**, 340–346 (1961).
- [23] Zerby, C. D., Meyer, A. & Murray, R. B. Intrinsic line broadening in NaI(Tl) gamma-ray spectrometers. *Nucl. Instrum. Methods* **12**, 115–123 (1961).
- [24] Hill, R. & Collinson, A. J. The relationships between light output and energy resolution in thallium activated sodium iodide crystals. *Nucl. Instrum. Methods* **44**, 245–252 (1966).
- [25] Prescott, J. R. & Narayan, G. H. Electron responses and intrinsic line-widths in NaI(Tl). *Nucl. Instrum. Methods* **75**, 51–55 (1969).
- [26] Dorenbos, P., de Haas, J. T. & van Eijk, C. W. Non-Proportionality in the Scintillation Response and the Energy Resolution Obtainable with Scintillation Crystals. *IEEE Trans. Nucl. Sci.* **42**, 2190–2202 (1995).
- [27] Valentine, J. D. The light yield nonproportionality component of scintillator energy resolution. *IEEE Trans. Nucl. Sci.* **45**, 512–517 (1998).

- [28] Moszyński, M. et al. Intrinsic energy resolution of NaI(Tl). *Nucl. Instrum. Methods Phys. Res. A* **484**, 259–269 (2002).
- [29] Valentine, J. D. & Rooney, B. D. Design of a Compton spectrometer experiment for studying scintillator non-linearity and intrinsic energy resolution. *Nucl. Instrum. Methods Phys. Res. A* **353**, 37–40 (1994).
- [30] Rooney, B. D. & Valentine, J. D. Scintillator light yield nonproportionality: Calculating photon response using measured electron response. *IEEE Trans. Nucl. Sci.* **44**, 509–516 (1997).
- [31] Dujardin, C. et al. Needs, trends, and advances in inorganic scintillators. *IEEE Trans. Nucl. Sci.* **65**, 1977–1997 (2018).
- [32] Narayan, G. H. & Prescott, J. R. The Contribution of the NaI(Tl) Crystal to the Total Linewidth of NaI(Tl) Scintillation Counters. *IEEE Trans. Nucl. Sci.* **15**, 162–166 (1968).
- [33] Mengesha, W. & Valentine, J. D. Benchmarking NaI(Tl) electron energy resolution measurements. *IEEE Trans. Nucl. Sci.* **49**, 2420–2426 (2002).
- [34] Świdorski, L. et al. Response of doped alkali iodides measured with gamma-ray absorption and Compton electrons. *Nucl. Instrum. Methods Phys. Res. A* **705**, 42–46 (2013).
- [35] Moszyński, M. et al. Energy resolution of scintillation detectors. *Nucl. Instrum. Methods Phys. Res. A* **805**, 25–35 (2016).
- [36] Breitenmoser, D., Butterweck, G., Kasprzak, M. M., Yukihiro, E. G. & Mayer, S. Laboratory based Spectral Measurement Data of the Swiss Airborne Gamma-ray Spectrometer RLL. ETH Research Collection <https://doi.org/10.3929/ethz-b-000528920> (2022).
- [37] Westmeier, W. Techniques and problems of low-level gamma-ray spectrometry. *Int. J. Rad. Appl. Instr. A* **43**, 305–322 (1992).
- [38] Coleman, T. F. & Li, Y. An Interior Trust Region Approach for Nonlinear Minimization Subject to Bounds. *SIAM J. Optim.* **6**, 418–445 (1996).
- [39] Rasmussen, C. E. & Williams, C. K. I. *Gaussian Processes for Machine Learning* (MIT Press, Cambridge, 2006).
- [40] Ahdida, C. et al. New Capabilities of the FLUKA Multi-Purpose Code. *Front. Phys.* **9**, 788253 (2022).

- [41] Payne, S. A. Nonproportionality of scintillator detectors. IV. Resolution contribution from delta-rays. *IEEE Trans. Nucl. Sci.* **62**, 372–380 (2015).
- [42] Breitenmoser, D., Cerutti, F., Butterweck, G., Kasprzak, M. M. & Mayer, S. FLUKA user routines for non-proportional scintillation simulations. ETH Research Collection <https://doi.org/10.3929/ETHZ-B-000595727> (2023).
- [43] Murray, R. B. & Meyer, A. Scintillation Response of Activated Inorganic Crystals to Various Charged Particles. *Phys. Rev.* **122**, 815–826 (1961).
- [44] Sobol, I. M. Global sensitivity indices for nonlinear mathematical models and their Monte Carlo estimates. *Math. Comput. Simul.* **55**, 271–280 (2001).

REVIEWER COMMENTS

Reviewer #1 (Remarks to the Author):

As discussed in my initial review, the work reported in this manuscript would advance the state of the art for scintillator characterization. Because inorganic scintillators are used in many fields, the work would be of broad interest to the scientific community and is, therefore, suitable for publication in Nature Communications. The methods are sound and ample detail is provided. The revised version compares results with previous work and discusses possible application of the methods beyond the scope of the current study. In my opinion, no further revisions are needed.

Reviewer #2 (Remarks to the Author):

On comment #1

The derivation and application of the semi-analytical model is impressive. The subsequent analysis definitively allows one to better understand the physics behind the observed shifts in energy. I thank the authors for adding the section S1.4 to the supplementary materials.

On comment #2

I thank the authors for the clarification.

On comment #3

I thank the authors for revealing that the sum mode inversion pipeline was the one presented in the submitted version of the manuscript and supplementary materials, and for adding information about the single mode inversion pipeline in the revised version of the manuscript and supplementary materials. As the authors mentioned, it is interesting to see that there are no statistically significant differences between the two pipelines, but still, understandably, observable differences between the crystals.

On comment #4

I thank the authors for sharing their thoughts on this subject, and I agree with them.

On comment #5

I thank the authors for the explanation.

On comment #6

I thank the authors for providing insights on the computational resources required for solving this type of problems. The time saved by using the surrogate model is indeed substantial, which certainly motivates its development and usage.

On comment #7

That's a good point. I thank the authors for this well thought and demonstrated justification.

On comment #8

I thank the authors for the explanation.

On comment #9

This is doubtlessly a great addition to the manuscript. The added details on the resolutions assuredly allow (1) a smoother reading, (2) a better understanding of the intrinsic resolution and of its relevance for this manuscript, and (3) to highlight the power and importance of the tool that the authors developed.

On comment #10

I thank the authors for the explanation and for elaborating on the energy range of applicability.

On comment #11

I thank the authors for adding the information.

On comment #12

Interesting. It is great to see that the development of the semi-analytical model led to a better understanding of the Compton edge shifts.

On comment #13

I thank the authors for including the pseudo-code in the supplementary materials, which answers the questions raised in this comment, and for reminding the reviewer about the ETH research data repository.

To conclude, I would like to thank the authors one more time for the time and efforts that they put towards the revision of the manuscript and supplementary materials. Everything was addressed far beyond expectations, which attests the excellence of this work.

Best of luck with this manuscript and with your future endeavors!

Best regards.

Reviewer #3 (Remarks to the Author):

Please see attached document, thanks.

Dear Researchers: Thank you for your thoughtful responses. I'd like to continue the conversation on one minor topic, with excerpts on my comment and your response below, with a follow-on from me.

(Original) Referee #3, Comment #3:

The comment "Nonproportionality effects are known to increase with crystal size" is not correct.

(Original) Response: We respectfully disagree with the referee that the statement "Nonproportionality effects are known to increase with crystal size" is incorrect. As correctly stated by the referee, the resolution or more specifically the intrinsic resolution, which is a direct consequence of the scintillation nonproportionality (cf. comment #9 for referee #2), increases with crystal size...

Second Response: I agree that a Compton scattering event followed by an additional Compton scatter will shift the edge to higher energy. But while nonproportionality could have some contribution to this effect, it is NOT required to generate the shift. All that is needed is two Compton events adding up to more than the single (maximum) Compton edge energy, and the release of a gamma photon from the scintillator. Consider the simplified gamma spectra below where nonproportionality is NOT included. On the left is the spectrum that includes Compton-followed-by-Compton events, while on the right the spectrum only includes Compton singles. The energy shift for the edge from multiple Compton scattering events is apparent – nonproportionality has a minor impact on it and is not required. I hope this makes sense... maybe think about it. I leave the final decision of how to handle this up to you.

Figure: (left) Spectrum with double Compton events; (right) Spectrum with only single Comptons.

5 Response to Referee #1

We would like to thank the referee once gain for the thoughtful feedback and positive evaluation. We believe that the modifications suggested by you have improved the manuscript significantly and made it more complete. We are really grateful and appreciate your time and effort in reviewing our manuscript. All the best for your future.

6 Response to Referee #2

We would like to thank the referee once again for the thorough review and the detailed feedback. We believe that the modifications suggested by you have improved the manuscript significantly and made it more complete. We are really grateful and appreciate your time and effort in reviewing our manuscript. All the best for your future.

7 Response to Referee #3

We thank the referee for pointing this out. We recognize that some of our statements in comment #3 in the last peer review report lacked clarity and might have been to a certain extent misleading. Furthermore, we agree that multiple (sequential or independent) Compton scatter events detected in coincidence in an inorganic scintillator such as NaI(Tl) might result in a spectral shift of the Compton edge toward higher spectral energies as highlighted in the figure provided by referee #3. That said, we believe that our statements and conclusions remain substantiated and are also in agreement with the referee's observations. In the following paragraphs, we provide a concise discussion of our findings and thereby offering a more in-depth explanation of our perspective on comment #3 from the previous peer review report.

First, it is important to note that we observe a shift of the Compton edge not toward higher but toward smaller spectral energies, i.e. a negative Compton edge shift (cf. Fig. 4 in the main manuscript as well as Fig. S26 in the Supplementary Information File). Based on these findings, we conclude that the process highlighted by the referee cannot be the sole factor contributing to the observed Compton edge shift.

Second, we have to underline that all our Monte Carlo simulations feature a high-fidelity fully coupled photon, electron and positron radiation transport as discussed in the Method section in the main manuscript (lines 374–376). Consequently, multiple Compton scatter events are properly simulated for the entire mass model in all performed simulations and the potential shift due to multiple Compton scatter events is included intrinsically in all our results. This is also true for the additional simulations described in the supplementary Section S1.4. Consequently, the process highlighted by the referee cannot be responsible for the observed discrepancy between measurements and simulations.

Third, as demonstrated in the main study in Section "Bayesian calibrated NPSM simulations" our NPSM predicts this negative Compton edge shift for all analyzed Compton edges with high accuracy. Moreover, with the semi-analytical model derived in the supplementary Section S1.4, we are able to show that the relation between photon energy, crystal size and negative Compton edge shift can be explained by the distribution of generated secondary electrons in the scintillator in combination with the decreasing trend in the relative light yield for higher energies (cf. supplementary Figs. S27–S29). These findings are in line with our statements in comment #3 in the previous peer review report. For more information on this topic, we kindly refer the reader to the dedicated supplementary Section S1.4 in the Supplementary Information File, which provides a detailed derivation and discussion of the results summarized here.

Motivated by the comment, we rephrased several statements in the main manuscript and the Supplementary Information File to enhance clarity and prevent potential misinterpretations by underlining the negative sign of the measured Compton edge shift. A full list of all minor changes made can be found in Section 8.

8 Summary of Changes

Here, we list all minor changes associated with the second peer review round.

1. In the main manuscript, we rephrased the sentence "In line with the Compton scatter theory (Supplementary Methods S1.4), we find a moderate increase in the Compton edge shift for higher photon energies." to "In line with the Compton scatter theory (Supplementary Methods S1.4), we find an enhanced shift of the Compton edge toward smaller spectral energies as the photon energy increases."
2. In the Supplementary Information File (p. S8), we rephrased the sentence "In general, we can identify a consistent trend in the shift, i.e. an increasing Compton edge shift with increasing Compton edge energy." to "In general, we can identify a consistent trend in the shift, i.e. an enhanced Compton edge shift toward smaller spectral energies with increasing Compton edge energy."
3. In the Supplementary Information File, we changed the phrase "Compton edge shift" to "negative Compton edge shift" at multiple locations in the supplementary Section S1.4, i.e. once in paragraph 6 on p. S9 and twice in the second last paragraph on p. S11.
4. In line with the guide for authors, we revised all vector-graphics, both in the main manuscript and the Supplementary Information File, to ensure that all display items are clipped at the corresponding axes outlines.

REVIEWER COMMENTS

Reviewer #3 (Remarks to the Author):

It appears that the issue was not a technical one, but a matter of better wording. I'm pleased that this issue has been cleared up. Thank you.

9 Response to Referee #3

We would like to thank the referee once again for the thorough review. We believe that the modifications motivated by your comments have improved the manuscript significantly and made it more complete. We are really grateful and appreciate your time and effort in reviewing our manuscript. All the best for your future.

10 Summary of Changes

Here, we list all minor changes associated with the third peer review round.

1. To avoid exaggerated language, we changed the following phrase in the last paragraph of the conclusion section in the revised manuscript from "The novel methodology presented in this study offers a reliable and cost-effective alternative ..." to "The methodology presented in this study offers a reliable and cost-effective alternative ...".
2. In accordance with the provided guide to prepare final artwork, we changed the label font size in all figures, both in the revised manuscript and the revised Supplementary Information File, from 8 pt to 7 pt. As a result, we also changed the tick font size in all figures from 7 pt to 6 pt.
3. In accordance with the provided guide to format the final article, we changed the font style of all mathematical vector terms from bold italic to bold non-italic in both, the revised manuscript and the Supplementary Information File (matrices were already in the correct format). In addition, we ensured that the vector font style in the revised figures is consistent with those changes in the manuscript and the Supplementary Information File.
4. In accordance with the provided guide to format the final article, we changed the symbol on line 20 on page S1 in the revised Supplementary Information File from ε_{PCA} to ε_{PCA} .
5. In accordance with the provided guide to format the final article, we rephrased the Data and Code availability statements in the revised manuscript to be in line with the provided template format.
6. In accordance with the provided guide to format the final article, we included a statement in the Acknowledgments of the revised manuscript highlighting the persons to which the declared funding was awarded.
7. In accordance with the provided guide to format the final article, we changed the title of the Supplementary Information File from "Supplementary Materials" to "Supplementary Information" as well as the titles of sections S1–S4 from "Materials & Methods", "Supplementary Figures", "Supplementary Tables" and "Supplementary Algorithms" to "Materials", "Figures", "Tables" and "Algorithms", respectively.
8. For consistency reasons, we changed the term "Supporting Information File" to "Supplementary Information File" in both, the revised manuscript and Supplementary Information File.